# Knowledge Entropy Decay during Language Model Pretraining Hinders New Knowledge Acquisition

**Jiyeon Kim**[1] *     **Hyunji Lee**[1] *     **Hyowon Cho**[1]     **Joel Jang**[2]

**Hyeonbin Hwang**[1]     **Seungpil Won**[3]     **Youbin Ahn**[3]     **Dohaeng Lee**[3]     **Minjoon Seo**[1]

[1] KAIST AI     [2] University of Washington     [3] LG AI Research

{jiyeon.kim, hyunji.amy.lee, minjoon}@kaist.ac.kr

## ABSTRACT

In this work, we investigate how a model's tendency to broadly integrate its parametric knowledge evolves throughout pretraining and how this behavior affects overall performance, particularly in terms of knowledge acquisition and forgetting. We introduce the concept of *knowledge entropy*, which quantifies the range of memory sources the model engages with; high knowledge entropy indicates that the model utilizes a wide range of memory sources, while low knowledge entropy suggests reliance on specific sources with greater certainty. Our analysis reveals a consistent decline in knowledge entropy as pretraining advances. We also find that the decline is closely associated with a reduction in the model's ability to acquire and retain knowledge, leading us to conclude that diminishing knowledge entropy (smaller number of active memory sources) impairs the model's knowledge acquisition and retention capabilities. We find further support for this by demonstrating that increasing the activity of inactive memory sources enhances the model's capacity for knowledge acquisition and retention.[1]

## 1 INTRODUCTION

Recent studies have analyzed how language models store world knowledge in their parameters and utilize this knowledge to generate responses during inference time (Geva et al., 2021; Dai et al., 2022a;b; Meng et al., 2022; Yao et al., 2024). However, little is known about how their behavior of integrating various factual knowledge embedded in their parameters changes throughout the pretraining stage. In this work, we perform a deep analysis of how a model's property of broadly integrating diverse parametric knowledge evolves throughout pretraining and how these shifts affect overall performance, particularly in terms of knowledge acquisition and forgetting in a continual learning setup. We hypothesize that this varying level of integration may explain why models in the later stages of pretraining encounter challenges in acquiring new knowledge (Dohare et al., 2024; Jang et al., 2022; Chang et al., 2024).

We introduce *knowledge entropy*, which reflects how a language model integrates various knowledge sources, to investigate how this behavior evolves throughout pretraining. Recent studies have shown that feed-forward layers (FFNs) serve as a key-value memory (Geva et al., 2022; 2021; Dai et al., 2022a). Building on this research, as shown in Figure 1, we view the second projection matrix $V$ as a memory, composed of memory

---

*Denotes equal contribution

[1]Code in https://github.com/kaistAI/Knowledge-Entropy.git

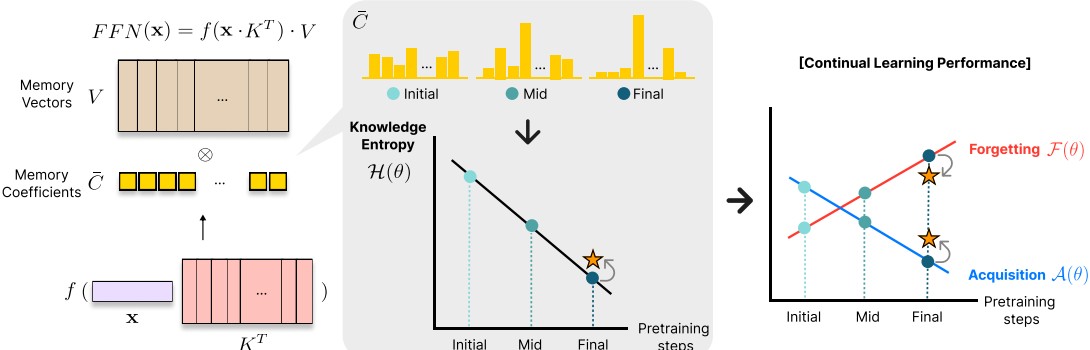

Figure 1: Illustration of our findings: distribution of memory coefficients $\bar{C}$ in feed-forward layers become sparser throughout pretraining, as indicated by a decrease in knowledge entropy $\mathcal{H}(\theta)$. This sparsity deteriorates the model's knowledge acquisition $\mathcal{A}(\theta)$ and increases forgetting $\mathcal{F}(\theta)$ when conducting continual knowledge learning with models from different pretraining stages. Thereby, as denoted by the star, when we artificially increase the knowledge entropy of the final stage model, both knowledge acquisition and retention increase.

vectors, which store the model's parametric knowledge, and view the first projection matrix $K$ as generating coefficients $\bar{C}$ that determine how these memory vectors are combined. Knowledge entropy measures how sparsely these memory coefficients are distributed; high knowledge entropy indicates that the model tends to integrate a broad range of memory vectors whereas the model with low knowledge entropy relies on specific memory vectors with high certainty. We analyze models at different stages of pretraining to investigate how knowledge entropy changes throughout pretraining. Our findings show that models in the later stages of pretraining tend to exhibit lower knowledge entropy, suggesting a shift from utilizing a larger set of active memory vectors to a smaller, more focused set as pretraining progresses.

We hypothesize that changes in knowledge entropy would influence the model's behavior when encountering new knowledge. To test this, we conduct a thorough analysis of the model's ability to acquire new knowledge and retain existing knowledge in a continual knowledge learning[2] scenario on a target corpus (Jang et al., 2022; Wu et al., 2023), starting from different stages of pretraining. This involves further training the pretrained model on new-domain corpora using a language modeling objective to integrate new knowledge. Our results reveal a strong correlation between knowledge entropy and the model's ability to acquire and retain knowledge: both knowledge entropy and knowledge acquisition and retention decrease as the pretraining progresses.

We assume that this correlation arises because lower knowledge entropy indicates a smaller set of active memory vectors, leading to frequent overwriting of these memory vectors to store new knowledge. To test this assumption, we conduct experiments where we artificially increase the activity of previously inactive memory vectors, allowing the model to store new knowledge across a broader range of memory vectors. Surprisingly, we observe that these modified models demonstrate improved knowledge acquisition and reduced forgetting compared to the original models when undergoing continual knowledge learning; though not to the same extent as the original pretrained model with equivalent knowledge entropy. Such a result bolsters our hypothesis that having a limited number of active memory vectors (low knowledge entropy) plays a critical role in explaining the degradation of the model's ability to acquire and retain knowledge as pretraining advances.

---

[2]In this work, we use the term "continual knowledge learning" and "continual learning" interchangeably.

Overall, our findings reveal that **as pretraining progresses, models exhibit a narrower integration of memory vectors, reflected by decreasing knowledge entropy, which hinders both knowledge acquisition and retention**. Models in the later stages of pretraining[3] show low knowledge entropy, leading to poor knowledge acquisition and higher forgetting rates despite being trained on larger datasets. In contrast, early-stage models display high knowledge entropy, enabling better knowledge acquisition and retention; however, their performance is often limited by weaker language modeling capabilities. Thus, mid-stage models strike a balance, showing strong knowledge acquisition and retention along with overall performance, making them a practical choice for further training to incorporate new knowledge. To the best of our knowledge, this is the first work to analyze how a model's behavior in integrating various memory vectors changes across pretraining stages and the subsequent effects on performance when acquiring new knowledge in a continual knowledge learning setup.

## 2 RELATED WORK

**Dynamics of Knowledge in Language Models**   Recent studies have shown that language models embed world knowledge within their parameters and integrate this knowledge to generate responses (Yang, 2024; Petroni et al., 2019; Wang et al., 2021). Thereby, various research efforts aim to understand these dynamics of knowledge in language models (how they learn, store, and engage their parametric knowledge) during inference and training phases. Several studies have focused on investigating the inference process: Geva et al. (2023) analyzes the role of different layers in language models. Allen-Zhu & Li (2024b) demonstrates that model parameters have a limited knowledge capacity. Some studies suggest key-value memory (Geva et al., 2021; Meng et al., 2022; Dai et al., 2022a). Other research focuses on the pretraining phase. Liu et al. (2021) studies the sequence that language models learn various types of knowledge. Allen-Zhu & Li (2024a) examines strategies to enhance knowledge storage and extraction. Teehan et al. analyzes internal structural changes. Sun & Dredze (2024) investigates the interaction between pretraining and finetuning. Chang et al. (2024) analyzes patterns of knowledge acquisition specifically during the pretraining process, addressing the question of how language models acquire knowledge during pretraining. While their study shares similarities with ours in investigating knowledge acquisition behavior during LLM training, our work takes a different focus. We aim to understand *why* LLMs encounter increasing difficulty in acquiring new knowledge as pretraining progresses, exploring the underlying reasons behind the challenges faced by later-stage models in learning new knowledge. To the best of our knowledge, our work is the first to explore how the behavior of language models in integrating their knowledge evolves throughout the pretraining phase, and to analyze how these changes affect model performance in terms of knowledge acquisition and forgetting in continual knowledge learning.

**Entropy in Natural Language Processing**   In information theory, entropy quantifies the value of information, where predictable (certain) events have low entropy and unpredictable (uncertain) events have high entropy (Lairez, 2022; Majenz, 2018). In natural language processing (NLP), entropy is used in various ways to measure the certainty of language models. Yang (2024) analyzes the entropy of model outputs based on input prompts. Araujo et al. (2022) calculates the entropy of outputs at each layer to determine weight adjustments in a continual learning setup. Other papers focus on token probability entropy to understand the information required to predict the next word in a sequence (Vazhentsev et al.; Geng et al., 2024; Malinin & Gales, 2021). Lower entropy in a model's predictions may indicate that the model has become more certain about its predictions based on training data. Additionally, Kumar & Sarawagi (2019) measures entropy over the cross-attention layer to assess the uncertainty in the attention layer of encoder-decoder models. The entropy proposed in our paper, *knowledge entropy*, differs from previous work in that it focuses on the

---

[3]We define the final stage as the last stage of the pre-determined learning rate schedule, with the most decayed learning rate. The mid-stage refers to around 50% of the schedule, while the initial stage refers to around 20%.

entropy of a model's parametric knowledge, assessing the uncertainty or variability in utilizing the knowledge encoded within the language model.

# 3 KNOWLEDGE ENTROPY

In this section, we introduce knowledge entropy (Section 3.1) to examine how broadly the model integrates its parametric knowledge and describe the experimental setup used to measure it (Section 3.2). Next, in Section 3.3, we measure knowledge entropy at various pretraining stages to analyze how the model's knowledge integration behavior evolves over pretraining. In Section 3.4, we extend our investigation by exploring alternative definitions of entropy.

## 3.1 DEFINITION

In this work, we introduce a new concept, *knowledge entropy*, to analyze the scope of a model's access patterns to its parametric knowledge. Low knowledge entropy suggests that the model relies on a narrower set of specific knowledge sources with high certainty whereas high knowledge entropy indicates that the model integrates with a diverse range of knowledge sources. Inspired by prior research that considers feed-forward layers (FFNs) as *key-value memory* containing a model's parametric memory (Geva et al., 2021; Dai et al., 2022a; Meng et al., 2022; Dong et al., 2022), we consider the knowledge source to be the *memory vectors*, which is the second projection matrix of FFN. We measure how broadly the model integrates these memory vectors with *memory coefficients*, which are calculated by the first projection matrix and the activation function.

Geva et al. (2021) propose the concept of *key-value memory*, demonstrating that FFNs function similarly to the key-value neural memories (Sukhbaatar et al., 2015). The feed-forward layer consists of two projection layers and activation in the middle:

$$FFN(\mathbf{x}) = f(\mathbf{x} \cdot K^T) \cdot V \tag{1}$$

where $\mathbf{x} \in \mathbb{R}^d$. The first projection matrix ($K \in \mathbb{R}^{m \times d}$) corresponds to the keys, and the second projection matrix ($V \in \mathbb{R}^{m \times d}$) represents the values, or the *memories* comprised of *memory vectors*. The output, $FFN(\mathbf{x})$, is a linear combination of the memory vectors $\mathbf{v}_{i=1,\cdots,m} \in \mathbb{R}^d$ which are the rows of $V$, where the *coefficients*[4] $C$ are determined by $f(\mathbf{x} \cdot K^T)$, with $f$ being a non-linear activation function such as ReLU. Previous studies have shown that various types of factual and linguistic knowledge are encoded within these memories (Dai et al., 2022a; Geva et al., 2022; Meng et al., 2022; Dong et al., 2022). Thus, the final output is generated by combining the contributions of these memory vectors, where the memory coefficients determine the combination.

Thereby knowledge entropy, $\mathcal{H}(\theta)$, is calculated by the sum of layer-wise entropy $\mathcal{H}(\theta^l)$, which is based on the average coefficient $\bar{C}^l \in \mathbb{R}^m$ averaged across all tokens in dataset $\mathcal{D}$, as described in Equation 2.

$$\bar{C}^l = \frac{1}{|\mathcal{D}|} \sum_{n=1}^{|\mathcal{D}|} \left( \frac{1}{T_n} \sum_{j=1}^{T_n} C_{n,j}^{(l)} \right); \qquad \text{prob}(\bar{c}_i^l) = \frac{\bar{c}_i^l}{\sum_{k=1}^{m} \bar{c}_k^l}, \quad \text{for } i = 1, 2, \ldots, m$$

$$\mathcal{H}(\theta^l) = -\sum_{i=1}^{m} \text{prob}(\bar{c}_i^l) \cdot \log(\text{prob}(\bar{c}_i^l)); \qquad \mathcal{H}(\theta) = \sum_{l=1}^{L} \mathcal{H}(\theta^l) \tag{2}$$

---

[4] We use the terms "coefficient" and "memory coefficient" interchangeably.

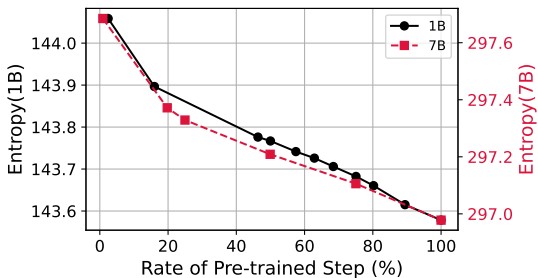 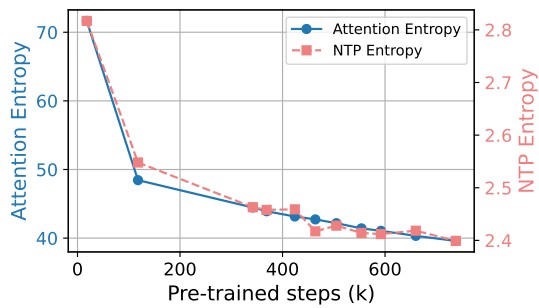

Figure 2: Entropy (y-axis) across different model states (x-axis) for OLMo 1B and 7B. The x-axis represents the rate of the current step relative to the last step (738k for 1B and 557k for 7B).

Figure 3: Entropy (y-axis) defined with Attention weight and next token prediction probability across different model states (x-axis) for OLMo 1B.

$C_{n,j}^{(l)}$ represents the coefficient of the $j$-th token position of the $n$-th instance at layer $l$, $\bar{c}_i^l$ indicates the $i$-th element from $\bar{C}^l$, $T_n$ is the sequence length of the $n$-th instance in the dataset $\mathcal{D}$, $m$ is the inner dimension of feed-forward layer, and $L$ denotes the number of layers in the model.

## 3.2 EXPERIMENT SETUP

To conduct the experiment, we use the OLMo (Groeneveld et al., 2024) models (1B and 7B), which are open-source large language models with intermediate pretraining checkpoints released, trained on the Dolma dataset (Soldaini et al., 2024)[5]. To measure knowledge entropy, we use a subset of Dolma, 2k instances that appear in the first batch within the official pretraining data order to ensure that all models we are using have seen the corpus during pretraining step. Please note that the trend persists across other corpora as well (Figure 7 in Appendix A.2); however, since we are analyzing the model's behavior throughout training, we define knowledge entropy based on calculations using the training dataset.

In the case of OLMo, the memory coefficient $C_{n,j}^{(l)}$ is calculated as $C_{n,j}^{(l)} = \text{abs}(\text{SwiGLU}(\mathbf{x_j}))$ where $\mathbf{x_j}$ is the j-th token of input $\mathbf{x}$ and SwiGLU (Shazeer, 2020) is the activation function. We apply the absolute value since the SwiGLU allows negative values and the magnitude determines the contribution of the corresponding memory vector in the linear combination. Then, the absolute values are converted into a probability distribution. We also show experimentally that the trend persists with different choices of activation functions. Further details regarding knowledge entropy can be found in Appendix A.2.

## 3.3 FINAL MODELS TEND TO EXHIBIT LOWER KNOWLEDGE ENTROPY

Figure 2 illustrates how knowledge entropy (y-axis) changes across different stages of pretraining (x-axis). The results show a **consistent decrease in knowledge entropy as pretraining progresses** in both 1B and 7B models. This trend suggests that models in the later stages of pretraining tend to engage with a narrower range of memories, relying more heavily on specific memory vectors rather than accessing and integrating knowledge from a broader range of memories. A consistent reduction in knowledge entropy is observed across all layers, with the most significant reduction occurring in the last layer, which closely resembles the output distribution right before the token prediction (Figure 8 in Appendix).

---

[5]We used OLMo and Dolma from the official repository

### 3.4 Similar Trends are Observed by Different Definitions of Entropy

While our work defines knowledge entropy focusing on the feed-forward layer, previous studies have examined entropy in different contexts, such as the entropy of attention (Kumar & Sarawagi, 2019) and the entropy of next-token prediction over the vocabulary space (Vazhentsev et al.; Malinin & Gales, 2021). To gain a more comprehensive understanding of the model's overall behavior, we extend our analysis by exploring the entropy trends in both attention mechanisms and next-token prediction. The formula and details are provided in Appendix A.3)

**Entropy of Attention Layers**    Following Kumar & Sarawagi (2019), we measure *attention entropy* to capture the degree of uncertainty in attention weights. It is calculated as the sum of layer-wise entropy, where the layer-wise entropy measures the sparsity of the attention weights in each attention head. Thus, *attention entropy* reflects how much weight the model assigns to specific tokens with confidence when generating the next token given the input based on token relationships. Figure 3 shows that attention entropy consistently decreases during pretraining, with a sharp decline in the early stages followed by a more gradual reduction. This trend suggests that the model learns to focus on contextually important tokens within the attention layer.

**Entropy of Next Token Prediction**    Entropy can also be measured based on the probability distribution of next-token predictions over the vocabulary space (Vazhentsev et al.; Geng et al., 2024; Malinin & Gales, 2021). Figure 3 shows that the entropy of the next token prediction also consistently decreases throughout pretraining, reflecting the model's increasing certainty in its next token prediction.

## 4 Knowledge Acquisition and Forgetting

We hypothesize that the reduction of knowledge entropy as pretraining progresses impacts the model's knowledge acquisition and forgetting as low knowledge entropy indicates sparse activation of memory vectors, thus the vectors are likely to be consistently overwritten when new knowledge is introduced. To test this hypothesis, we measure knowledge acquisition and forgetting using checkpoints from different stages of pretraining in a continual knowledge learning setup (Jang et al., 2022; Wu et al., 2023), where further training is performed on new-domain corpora by next token prediction to inject new knowledge into the pretrained models. Section 4.1 details the experimental setup and the metrics used. In Section 4.2, we present the results of knowledge acquisition and forgetting across various pretraining stages. Finally, Section 4.3 further explores whether a relationship exists between the two behaviors: activating the inactive memory vectors increases the knowledge acquisition ability.

### 4.1 Experiment Setup

**Model & Hyperparameters**    We experiment using intermediate checkpoints from OLMo[6]. Hyperparameters are chosen following previous research on continual knowledge learning (Jang et al., 2022; Kim et al., 2023) and we test various combinations to assess their generalizability. For batch size, we test 128 and 2048; for learning rate, we experiment with 1e-4, 4e-4, and 1e-3. We also investigate the effect of training duration by comparing a single epoch to three epochs. Among these configurations, we focus primarily on a batch size of 128, a learning rate of 4e-4, and single-epoch training as this setup most closely aligns with continual knowledge learning.

---

[6]We select these intermediate checkpoints following Sun & Dredze (2024), which explores language model (OLMo) throughout pretraining. Available checkpoints can be found here.

**Dataset** We experiment on a subset of two datasets[7]: PubMed [8], a corpus of bio-medical and life science topics with abstracts, and C4 (Raffel et al., 2020), a large-scale corpus comprising diverse text data gathered from web pages. We use PubMed as the primary dataset as it contains more new knowledge, making it a better fit for our continual knowledge learning setup (Appendix B.1). In addition to the dataset, we inject synthetic knowledge during training to assess the model's ability to acquire new information. Specifically, we utilize FICTIONAL KNOWLEDGE dataset (Chang et al., 2024), which is designed to assess how well language models acquire factual knowledge during pretraining[9]. This dataset includes 130 paragraphs about fictional yet realistic entities and 1,950 probes where each paragraph contains 15 different probes. The passages are incorporated into the training batch 10 times during the continual knowledge learning. After training, we evaluate the models on evaluation probes of the Fictional Knowledge dataset to measure knowledge acquisition, and evaluate on six downstream tasks in zero-shot manner (Sun & Dredze, 2024; Groeneveld et al., 2024) to measure knowledge forgetting (SciQ (Welbl et al., 2017), Winogrande (Sakaguchi et al., 2021), PIQA (Bisk et al., 2020), OBQA (Mihaylov et al., 2018), HellaSwag (Zellers et al., 2019), and ARC Easy (Clark et al., 2018)). Detailed explanation is included in Appendix B.1.

**Metric** Knowledge acquisition of a language model $\theta$ is measured with the probing performance on evaluation probes following Chang et al. (2024). When given the injected knowledge $W$, each instance $w_i$ in a corpus has a corresponding set of probes $P_{w_i}$, containing 15 different probes. To measure how well the model recalls the injected knowledge, we compute the **probe performance** $\mathcal{K}(\theta)$, the average log probability $\ell(p_i;\theta)$ of the target span for each probe $p_i \in P_{w_i}$ across all instances in $w_i \in W$ and calculate average; $\mathcal{K}(\theta) = \frac{1}{|W|} \sum_{w_i \in W} \frac{1}{|P_{w_i}|} \sum_{p_i \in P_{w_i}} \ell(p_i;\theta)$. The **knowledge acquisition metric** $\mathcal{A}(\theta)$ is defined as the improvement rate of $\mathcal{K}(\theta)$ from $\theta_{\text{PT}}$ to $\theta_{\text{CL}}$, where $\theta_{\text{PT}}$ represents the model checkpoint from a pretraining step, which serves as the starting point and $\theta_{\text{CL}}$ represents the model after continual knowledge learning. High $\mathcal{A}(\theta)$ indicates the model has learned new knowledge well.

To measure knowledge forgetting of a language model, we measure **average performance over six downstream tasks** $\mathcal{P}(\theta)$. **Knowledge forgetting** $\mathcal{F}(\theta)$ is calculated by the reduction rate from $\theta_{\text{PT}}$ to $\theta_{\text{CL}}$. Low $\mathcal{F}(\theta)$ indicates that the models have retained their existing knowledge. The equation and a detailed explanation are presented in Appendix B.2.

### 4.2 KNOWLEDGE ACQUISITION AND RETENTION DECREASES ACROSS PRETRAINING STAGE

Figure 4a shows the performance of OLMo 1B and 7B models [10] from various stages of pretraining as an initial state. **We observe that models in the final stages of pretraining struggle more with acquiring new knowledge** $\mathcal{A}(\theta)$ **and exhibit greater forgetting** $\mathcal{F}(\theta)$. As shown in Figure 4b, continually training the models at the mid-point of the pretraining as the initial checkpoint tends to yield the best performance in knowledge probing and downstream tasks compared to both models from the initial and final stages of pretraining. While early-stage models demonstrate high knowledge acquisition with minimal forgetting, their overall performance is limited by weaker language modeling capabilities. Conversely, later-stage models, despite being trained on larger datasets, exhibit lower knowledge acquisition and higher rates of forgetting, resulting in lower overall performance compared to the mid-stage models. This aligns with previous research suggesting that a model in the final stage of pretraining tends to struggle when learning new knowledge, showing a trade-off between *plasticity* and *stability* (Dohare et al., 2024; Biesialska et al., 2020; Jang et al.,

---

[7]We randomly sample 205k instances for each dataset.

[8]Datasets in huggingface

[9]We slightly modified the dataset to our setting of which details are in Appendix B.1

[10]For this experiment, we used PubMed as a training corpus, but we also experiment with C4 as a training corpus, which presumably exhibits more similar distribution to the pretraining corpus, Dolma. The comparison of the two corpora and the results can be found in Appendix B.1 and B.4.2

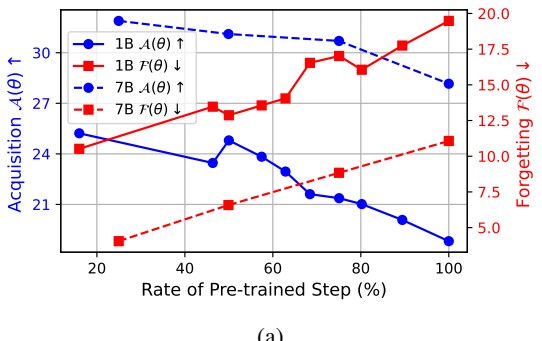 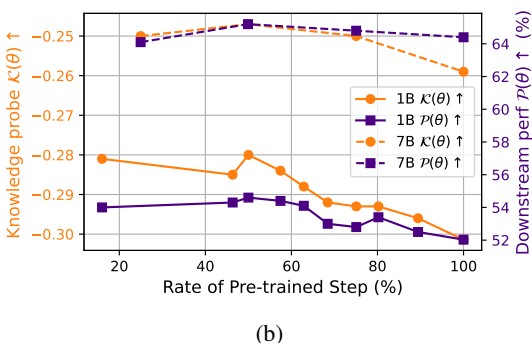

(a)                                                      (b)

Figure 4: (a) Rates of knowledge acquisition $\mathcal{A}(\theta)$ and forgetting $\mathcal{F}(\theta)$ (b) End performance of knowledge probe $\mathcal{K}(\theta)$ and downstream task $\mathcal{P}(\theta)$ for OLMo 1B (solid line) and 7B (dotted line) across different model states. The x-axis represents the ratio of the pretraining step of the initial model used in continual learning to the final step. (738k for 1B and 557k for 7B)

2022). Therefore, we suggest that using a mid-stage checkpoint strikes a good balance, making it a practical choice as an initial starting point for further training to inject new knowledge.

We consistently observe this pattern of later-stage models underperforming compared to earlier-stage models across various hyperparameter settings, including batch size, learning rate, training corpus, and the number of epochs. A detailed analysis of these results is provided in Appendix B.4.

### 4.3 RESUSCITATING INACTIVE MEMORY VECTORS INCREASES KNOWLEDGE ACQUISITION

We observe a strong correlation[11] between the trend of knowledge entropy (Figure 2) and the model's ability to acquire and retain knowledge (Figure 4a). We assume that the model's increasing reliance on a limited set of memory vectors (as indicated by a decrease in knowledge entropy) leads to more frequent updates to these vectors, making it difficult to acquire new knowledge and resulting in a higher rate of forgetting. (The intuition behind this hypothesis can be found in Appendix A.1) To test this assumption, we conduct experiments where we artificially increase the activity or resuscitate previously inactive memory vectors.

To resuscitate inactive memory vectors, we modify the up-projection matrix $K$ which engages with producing memory coefficients $\bar{C}$ (notations from Equation 2). Specifically, as shown in Algorithm 1, we identify the lowest $p\%$ (resuscitation ratio) of memory coefficients and apply a multiplier $u$ to parameters in $K$ that are associated with these $p\%$. Multiplier $u$ can take any value; in this experiment, we divide the mean coefficient value of each layer by the respective coefficient value $\bar{c}_i^l$ at each identified position $i$ at layer $l$, and then multiply the result by an amplifying factor $q$. By varying the value of $q$, we control the degree of resuscitation applied to the $p\%$ low-activation coefficients, thereby influencing the magnitude of the average coefficient and the corresponding size of the parameter updates.

Figure 5a shows the knowledge acquisition and forgetting rates and Figure 5b presents the knowledge probe and downstream task performance after continual learning with various resuscitation configurations. For the experiment, we fix $p$ to 50 with varying $q$ and use the OLMo checkpoint at the last step of pretraining. Results show that when $q$ is set to 1 or greater, it generally yields better performance in both knowledge acquisition and retention compared to the original model. In contrast, when $q$ is set to 0.5, which further reduces already

---

[11]The Pearson correlation between knowledge entropy and knowledge acquisition is 0.94, and with forgetting, it is -0.96. Both correlations are statistically significant, with p-values of 6e-5 and 1e-5, respectively.

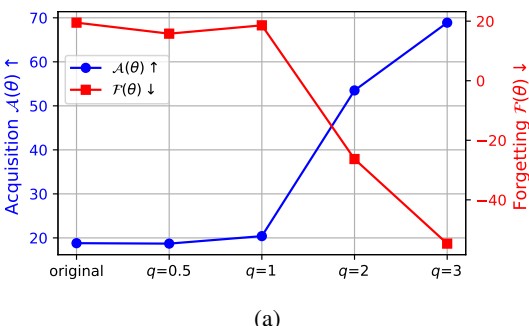 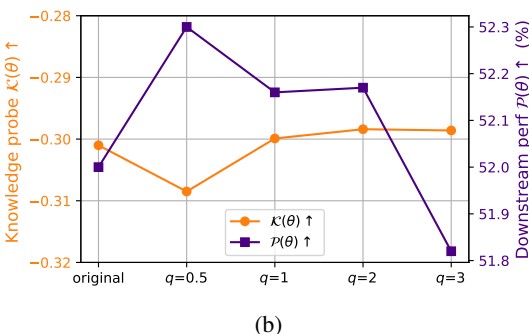

(a)                                                    (b)

Figure 5: (a) Rates of knowledge acquisition $\mathcal{A}(\theta)$ and forgetting $\mathcal{F}(\theta)$ (b) End performance of knowledge probe $\mathcal{K}(\theta)$ and downstream task $\mathcal{P}(\theta)$ after training with varying the amplifying factor $q$ (x-axis) while $p$ is fixed as 50.

inactive memory coefficients, both acquisition and retention decline suggesting that concentrating parameter updates more heavily on already active locations leads to sparser updates, ultimately impairing overall performance. These results suggest that **having a narrower active memory vector (low knowledge entropy) tends to reduce the model's capacity to acquire new knowledge and increases knowledge forgetting**.

Further experiments with fixed $q$ and varying $p$ show that increasing $p$ to activate a larger portion of inactive parameters generally leads to improved performance. Detailed result of this configuration is in Appendix B.6. We also analyze how the result changes when using models from different stages of pretraining as the original model. The trend holds consistently across different checkpoints of pretraining. However, the effect of the resuscitation becomes more pronounced as the original model progresses to later stages of pretraining. Detailed results are included in Appendix B.7.

Our result indicates that resuscitating inactive memory vectors of final-stage models tends to enhance knowledge acquisition and overall performance compared to the unmodified final-stage model. However, we observe that the performance remains lower compared to models from the pretraining step with similar knowledge entropy, such as the mid-stage model. This suggests that applying linear scaling to a subset of specific layers alone is insufficient to induce fundamental behavioral changes in the model. In other words, to restore a model that has lost its *plasticity* (Dohare et al., 2024) to its previous state, more fundamental and alternative approaches are required. Further exploring methods for effectively modifying the parameters would be an interesting direction for future work.

## 5    CONCLUSION

In this work, we examine how large language models' ability to broadly integrate their parametric knowledge (measured by knowledge entropy) changes throughout pretraining and how these changes affect knowledge acquisition and forgetting in a continual learning setup. Our findings reveal a strong correlation between knowledge entropy and the model's capacity to acquire and retain knowledge. Models in the final stages of pretraining tend to exhibit narrower integration of memory vectors, leading to lower knowledge entropy, which negatively impacts both knowledge acquisition and retention. Interestingly, artificially increasing knowledge entropy by modifying the parameters of final-stage models tends to improve these capabilities. Based on our analysis, we suggest that models from the mid-stage of pretraining offer a good balance between knowledge acquisition, retention, and overall performance, making them a good choice for further training to introduce new knowledge.

---

**Algorithm 1** Resuscitating Low Memory Coefficients

---

**Require:** $\bar{C}$ (average coefficients), $p$ (resuscitation ratio), $q$ (amplifying factor), $K$ (up-projection matrix)
**Ensure:** Scaled up-projection matrix $K$ using computed multiplier $u$
 1: **for** each layer $l$ in $K$ **do**
 2:     Extract average activations for layer $l$:
$$C = \bar{C}[l]$$
 3:     Compute the threshold $t$ for the lowest $p\%$ activations:
$$t = \text{percentile}(C, p)$$
 4:     Identify positions of values below the threshold $t$:
$$\text{idx} = (C \leq t).\text{nonzero}(\ )$$
 5:     Compute the multiplier $u$ for coefficients in layer $l$:
$$u = \frac{\text{mean}(C)}{C} \times q$$
 6:     Apply scaling to the up-projection weights $K_l$ at the identified positions:
$$K_l[\text{idx}, :] \ \times= \ u[\text{idx}]$$
 7: **end for**

---

## 6    LIMITATION & FUTURE WORK

Due to computational constraints, our study measures knowledge acquisition and forgetting in a continual learning setup. Future work could explore whether these behaviors also manifest during the pretraining phase. We focused on OLMo 1B and 7B models, as they are the only models that publicly provide intermediate pretraining checkpoints and demonstrate strong performance (Sun & Dredze, 2024; Chang et al., 2024). Extending this investigation to other models would be a valuable direction for further research. Our resuscitation method, which arbitrarily modifies model parameters to test our hypothesis, showed promising results in improving knowledge acquisition and retention. However, performance tended to decline when resuscitating models at their initial or mid-stages. This suggests that more refined methods for resuscitating model parameters—ones that avoid random modification and preserve language modeling capabilities—could yield better outcomes. Additionally, while we observed that models in the mid-stage of pretraining strike a good balance for further training on tasks that involve acquiring new knowledge, defining the mid-point precisely remains an open question. In this study, we approximated the mid-point as 50% of the learning rate schedule.

ACKNOWLEDGMENTS

We thank Sohee Yang, Hoyeon Chang, Seongyun Lee, Seungone Kim, Doyoung Kim, and Hanseok Oh for helpful discussions and constructive feedback.

This work was supported by LG AI Research grant (Self-improving logical reasoning capabilities of LLMs, 2024, 50%) and the Institute of Information & Communications Technology Planning & Evaluation(IITP) grant funded by the Korea government(MSIT) (RS-2024-00397966, Development of a Cybersecurity Specialized RAG-based sLLM Model for Suppressing Gen-AI Malfunctions and Construction of a Publicly Demonstration Platform, 20%; No.2022-0-00113, Developing a Sustainable Collaborative Multi-modal Lifelong Learning Framework, 20%; No.RS-2021-II212068, Artificial Intelligence Innovation Hub, 10%)

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

## A   KNOWLEDGE ENTROPY

### A.1   INTUITION BEHIND THE DEFINITION OF KNOWLEDGE ENTROPY

The mechanism behind the relationship between decreasing knowledge entropy and the ability to acquire and retain knowledge during pretraining is that the coefficients in the linear combination of memory vectors determine how the corresponding memory vectors are updated. As defined in Equation 1, the output of the feed-forward layers (FFNs) is a linear combination of memory vectors $\mathbf{v}_{i=1,\cdots,m} \in \mathbb{R}^d$, the row vectors of $V \in \mathbb{R}^{m \times d}$, where the coefficients $c_i$ are given by $f(\mathbf{x} \cdot K^T)$. In other words, $FFN(x) = c_1\mathbf{v_1} + c_2\mathbf{v_2} + \cdots + c_m\mathbf{v_m}$. Within a given layer, since the operations beyond the FFNs and the input to the FFNs remain consistent across all memory vectors, the coefficients act as scaling factors for the gradient by the chain rule. During training, the gradient $\frac{\partial L}{\partial \mathbf{v_{i,j}}}$ for $i = 1, 2, \ldots, m$ and $j = 1, 2, \ldots, d$ can be decomposed as:

$$\frac{\partial L}{\partial \mathbf{v_{i,j}}} = \frac{\partial L}{\partial FFN(x)} \cdot \frac{\partial FFN(x)}{\partial v_{i,j}}$$

Here, $\frac{\partial L}{\partial FFN(x)}$ is the same for all $\mathbf{v_i}$, meaning that the relative magnitude of the gradient depends on $\frac{\partial FFN(x)}{\partial v_{i,j}} = c_i$. Thus, larger coefficients result in proportionally larger gradients being applied to the corresponding memory vectors, amplifying their updates during backpropagation.

As pretraining progresses, a spikier coefficient distribution—captured by decreasing knowledge entropy—implies that gradient updates become increasingly concentrated on specific positions where the average coefficients are larger. This centralization can affect the model's ability to evenly utilize its memory capacity, thereby impacting knowledge acquisition and retention.

### A.2   KNOWLEDGE ENTROPY

**Does the choice of model change the trend?**   To assess the generalizability of the trend observed in Figure 2, we conducted experiments on the knowledge entropy trend using Pythia 1.4B model (Biderman et al., 2023). As shown in Figure 6, knowledge entropy measured with the Pythia model also tends to decrease as pretraining progresses.

**Does the choice of dataset change the trend?**   As expressed in Equation 2, knowledge entropy is dependent on the dataset $\mathcal{D}$. We define $\mathcal{D}$ as the dataset used during pretraining, as knowledge entropy reflects how the model integrates the knowledge stored in its memory vectors, learned during pretraining. However, to further explore whether the choice of dataset influences the trend of knowledge entropy, we measure it using PubMed and C4. Figure 7 shows that the trend remains consistent regardless of the dataset used when calculating knowledge entropy.

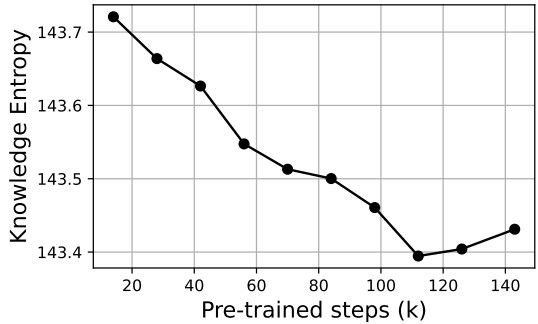

Figure 6: Knowledge Entropy (y-axis) measured with Dolma on different pretraining stages of Pythia 1.4B model.

Figure 7: Knowledge Entropy(y-axis) measured with Dolma, C4, Pubmed corpus across different pretraining stages

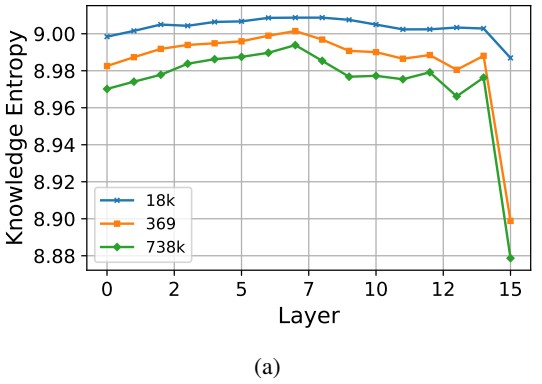

(a)

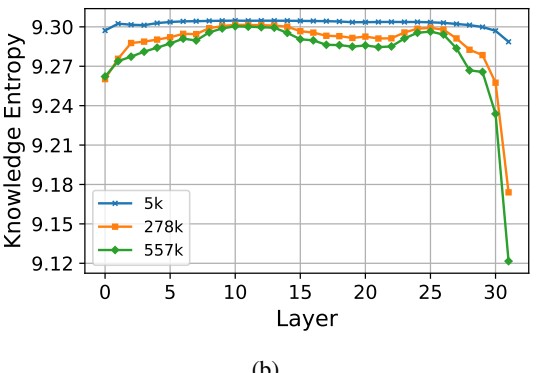

(b)

Figure 8: Layer-wise Knowledge Entropy of OLMo-1B(a) and 7B(b)

**Does the choice of activation function change the trend?**    We also explored an alternative where we do not take the absolute value of the SwiGLU output. Instead, following the ReLU function (Agarap, 2018), another widely used activation function, we replaced all negative values with 0. Figure 9 shows that the trend remains consistent even under this modification.

$$C_{n,j}^{(l)} = \text{ReLU}(\text{gate}(\mathbf{x_j})) \otimes \text{up}(\mathbf{x_j}),$$

**Layer-wise Knowledge Entropy**    Figure 8 shows how knowledge entropy changes by layer during pretraining. Knowledge entropy consistently decreases in every layer, with the most significant reduction occurring in the last layer, which closely resembles the output distribution right before the token prediction. OLMo-7B model also shows a similar trend to 1B model.

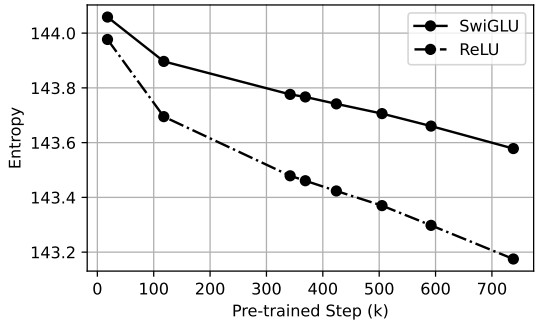

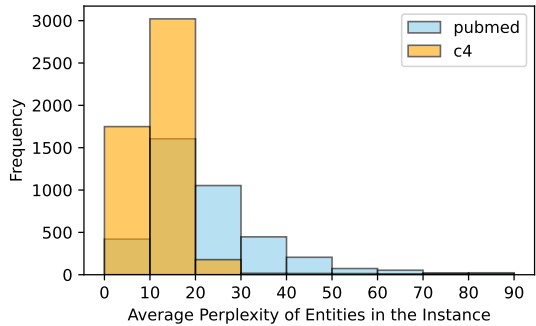

Figure 9: Knowledge Entropy(y-axis) when employing original SwiGLU(solid line) and ReLU activation(dotted line) across different pretraining stages

Figure 10: Histogram of average perplexity of entities in C4 and Pubmed corpus

### A.3 ENTROPY OF ATTENTION LAYERS

Inspired by previous research that emphasizes the attention layer's role of attribute extraction (Geva et al., 2023), we also measure *attention entropy* $\mathcal{H}(\theta_{\text{att}})$ similarly to Kumar & Sarawagi (2019)[12]. Attention weights, which are the output of softmax normalization after the key-query-value operation, can be interpreted as weight assigned to the previous tokens. As the attention weight for each token position in each attention head forms a probability distribution(summing to 1), calculating entropy follows the normal entropy formula. Then, layer-wise entropy $\mathcal{H}(\theta_{\text{att}}^l)$ is averaged over token position and attention heads and attention entropy $\mathcal{H}(\theta_{\text{att}})$ is the sum of layer-wise entropy $\mathcal{H}(\theta_{\text{att}}^l)$. Following the notations from Geva et al. (2023), attention entropy is calculated as:

$$\mathcal{H}^{n,j}(\theta_{\text{att}}^{h,l}) = -\sum_{i=1}^{j} \mathbf{A}_{i,j}^{h,l,n} \cdot \log(\mathbf{A}_{i,j}^{h,l,n}) \quad \text{for } i = 1, 2, \ldots, T_n \text{ and } j = 1, 2, \ldots, T_n$$

$$\mathcal{H}(\theta_{\text{att}}^{h,l}) = \frac{1}{|\mathcal{D}|}\sum_{n=1}^{|\mathcal{D}|}\left(\frac{1}{T_n}\sum_{j=1}^{T_n}\mathcal{H}^{n,j}(\theta_{\text{att}}^{h,l})\right); \quad \mathcal{H}(\theta_{\text{att}}^l) = \frac{1}{N}\sum_{h=1}^{N}\mathcal{H}(\theta_{\text{att}}^{h,l}); \quad \mathcal{H}(\theta_{\text{att}}) = \sum_{l=1}^{L}\mathcal{H}(\theta_{\text{att}}^l)$$

(3)

where $\mathbf{A}^{h,l,n} \in \mathbb{R}^{T_n \times T_n}$ represents the attention weights of the $h$-th attention head in layer $l$ for the $n$-th instance, $T_n$ is the sequence length of the $n$-th instance in the training dataset $\mathcal{D}$, N denotes the number of attention heads, and $L$ denotes the number of layers in the model.

### A.4 ENTROPY OF NEXT TOKEN PREDICTION

The entropy of next token prediction (Vazhentsev et al.; Geng et al., 2024; Malinin & Gales, 2021) is defined as $\mathcal{H}(\theta_{\text{ntp}}^{n,j}) = -\sum_{i=1}^{|V|} p_i \cdot \log(p_i)$, where $p_i$ represents the probability of the $i$-th token. This value is then averaged over the sequence length ($T_n$) and the dataset size ($|\mathcal{D}|$).

---

[12]Kumar & Sarawagi (2019) measures entropy using attention weights from cross-attention in an encoder-decoder architecture. However, as we employ a decoder-only model, we modify the equation to use attention weights from self-attention.

## B  KNOWLEDGE ACQUISITION AND FORGETTING

### B.1  DATASETS

**Training Dataset for Continual Knowledge Learning**    In this section, we share a brief description of the datasets we used. For continual knowledge learning, we experiment with PubMed and C4. The PubMed dataset consists of biomedical literature abstracts from the PubMed database, containing articles across a wide range of topics in medicine and biology. The C4 (Colossal Clean Crawled Corpus) dataset (Raffel et al., 2020) is a large-scale, preprocessed collection of text scraped from the web, designed to be a clean and diverse representation of natural language.

To compare the distribution of C4, PubMed, and Dolma, we evaluate the average perplexity of entities for C4 and PubMed, as shown in Figure 10. On the x-axis, we plot the range of an average perplexity of instances, while the y-axis represents the number of instances. We randomly sample 10,000 instances from each corpus, extract entities using GPT-4o, and calculate perplexity with the last checkpoint of OLMo. The perplexity values from the last checkpoint of OLMo indicate how likely these entities are to appear in the pretraining corpus, Dolma. The results reveal that PubMed exhibits a broad distribution of perplexity, with a higher number of instances having high perplexity values. In contrast, C4 shows a tendency towards lower perplexity, suggesting that the distribution of entities in PubMed differs from that in Dolma, while the distribution in C4 tends to be more similar to Dolma.

**Evaluation Dataset to measure *Knowledge Acquisition***    To measure knowledge acquisition of language model, we use the fictional knowledge dataset (Chang et al., 2024), which is designed to assess how well LLMs acquire factual knowledge during pretraining. This dataset includes 130 paragraphs [13] presented in a Wikipedia-style format with fictional yet realistic entities (injected knowledge) and 1,950 probes which are cloze-task-style sentences to query the information within the corpus. The final span of each probe, referred to as the target span, is used to evaluate the model's prediction probability, which serves as a measure of knowledge acquisition performance.

In Chang et al. (2024), the probes are divided into three levels of difficulty, with five sentences created for each level. This results in 15 probes per corpus. The difficulty levels are as follows: 1) **Memorization probes** directly ask about sentences explicitly present in the fictional corpus. 2) **Semantic generalization probes** are paraphrased versions of the memorization probes to test the model's understanding of meaning beyond surface forms. 3) **Compositional generalization probes** are designed to assess whether the model can integrate multiple pieces of knowledge from the fictional corpus. The injected knowledge is incorporated into the training corpus during continual learning, with updates occurring every 160 steps.

Following Chang et al. (2024), we divide the 130 corpora into two settings: *paraphrase* and *once*. In the *paraphrase* setting, 70 instances are each paraphrased 10 times. For every 160 steps, one paraphrased version of an instance is added to the training corpus, repeating this process 10 times[14]. In the *once* setting, each instance is presented only once throughout the entire continual learning process. The 60 instances are divided into 10 groups, with 6 instances added every 160 steps.

**Evaluation Dataset to measure *Knowledge Forgetting***    To measure the forgetting rate, we evaluate on 6 downstream datasets.

---

[13]While Chang et al. (2024) utilizes 120 paragraphs, comprising 40 for *paraphrase* setup, 40 for *duplicate*, and 40 for *once*, we utilized 130 paragraphs, the whole original data. We scaled up the number of paragraphs with paraphrases to 70, utilizing GPT-4 following Chang et al. (2024)

[14]We maintain updates every 160 steps(10 steps when batch size is 2048) because our total training duration is 1,600 steps(100 steps), and we repeat the dataset injection process 10 times.

- SCIQ: multiple-choice question-answering dataset consisting of over 13,000 science exam-style questions, covering subjects such as physics, chemistry, biology, and earth science

- WINOGRANDE: large-scale benchmark designed to test commonsense reasoning in natural language understanding

- PIQA: commonsense reasoning about everyday physical interactions such as how to perform tasks involving physical actions

- OBQA: multiple-choice question-answering benchmark designed to assess a model's ability to answer elementary-level science questions.

- HELLASWAG: a large-scale benchmark for commonsense reasoning, focusing on selecting the most plausible continuation of a given narrative or scene.

- ARC EASY: multiple-choice science questions typically answered by students in elementary and middle school.

## B.2 METRIC

In this section, we share a detailed description of how we evaluate *knowledge acquisition* and *knowledge forgetting*.

**Knowledge Acquisition**  Given a language model $\theta$, $\theta_{\mathrm{PT}}$ represents the model extracted from a pretraining step and serves as the initial point for continual learning, and $\theta_{\mathrm{CL}}$ represents the model after it. The acquisition metric $\mathcal{A}(\theta)$ for the model $\theta$ is defined as Equation 4. When given a corpus set of *once* setting $W_{\mathrm{once}}$, each instance in a corpus $w_i$ has a corresponding set of probes $P_{w_i}$, which contains 15 different probes. To calculate the performance of the *once* setting, $\mathcal{K}_{\mathrm{once}}(\theta)$, we compute the average log probability $\ell(p_i;\theta)$ of the target span for each probe $p_i \in P_{w_i}$ across all instances in $w_i \in W_{\mathrm{once}}$ and sum these averages. The same calculation is performed for the *paraphrase* setting. The total performance $\mathcal{K}(\theta)$ is calculated by the weighted average of $\mathcal{K}_{\mathrm{once}}(\theta)$ and $\mathcal{K}_{\mathrm{para}}(\theta)$. Finally, the acquisition metric $\mathcal{A}(\theta)$ is defined as the improvement rate in performance $\mathcal{K}(\theta)$ from the initial model state $\theta_{\mathrm{PT}}$ to final model state $\theta_{\mathrm{CL}}$.

$$\mathcal{K}_{\mathrm{once}}(\theta) = \frac{1}{|W_{\mathrm{once}}|} \sum_{w_i \in W_{\mathrm{once}}} \frac{1}{|P_{w_i}|} \sum_{p_i \in P_{w_i}} \ell(p_i;\theta); \qquad \mathcal{K}_{\mathrm{para}}(\theta) = \frac{1}{|W_{\mathrm{para}}|} \sum_{w_i \in W_{\mathrm{para}}} \frac{1}{|P_{w_i}|} \sum_{p_i \in P_{w_i}} \ell(p_i;\theta)$$

$$\mathcal{K}(\theta) = \frac{|W_{\mathrm{once}}| \times \mathcal{K}_{\mathrm{once}}(\theta) + |W_{\mathrm{para}}| \times \mathcal{K}_{\mathrm{para}}(\theta)}{|W_{\mathrm{once}}| + |W_{\mathrm{para}}|}; \qquad \mathcal{A}(\theta) = \frac{\mathcal{K}(\theta_{\mathrm{CL}}) - \mathcal{K}(\theta_{\mathrm{PT}})}{\mathcal{K}(\theta_{\mathrm{PT}})} \tag{4}$$

**Knowledge Forgetting**  The forgetting metric $\mathcal{F}(\theta)$ is calculated as the average performance degradation from the initial model $\theta_{\mathrm{PT}}$ to the final model $\theta_{\mathrm{CL}}$ across six downstream tasks $\mathcal{T}$: SciQ (Welbl et al., 2017), Winograde (Sakaguchi et al., 2021), PIQA (Bisk et al., 2020), OBQA (Mihaylov et al., 2018), HellaSwag (Zellers et al., 2019), and ARC Easy (Clark et al., 2018).

$$\mathcal{P}(\theta) = \frac{1}{|\mathcal{T}|} \sum_{i \in |\mathcal{T}|} \mathcal{T}_i(\theta); \qquad \mathcal{F}(\theta) = -\frac{\mathcal{P}(\theta_{\mathrm{CL}}) - \mathcal{P}(\theta_{\mathrm{PT}})}{\mathcal{P}(\theta_{\mathrm{PT}})} \tag{5}$$

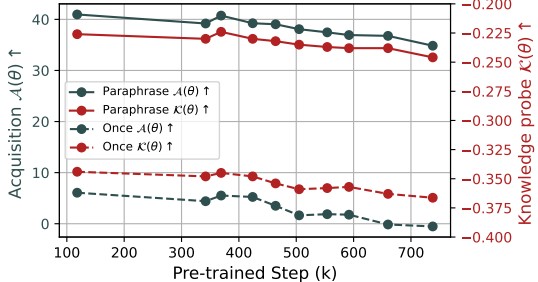

Figure 11: Rates of knowledge acquisition $\mathcal{A}(\theta)$ and final performance of knowledge probe $\mathcal{K}(\theta)$ according to the number of injection. Paraphrase prompts (solid line) were injected 10 times and Once (dotted line) prompt only once throughout continual training.

Figure 12: End performance of knowledge probe $\mathcal{K}(\theta)$ and downstream task $\mathcal{P}(\theta)$ of initial model (dotted line) and model after continual training (solid line) in baseline setup. The x-axis represents the initial model state used for continual training.

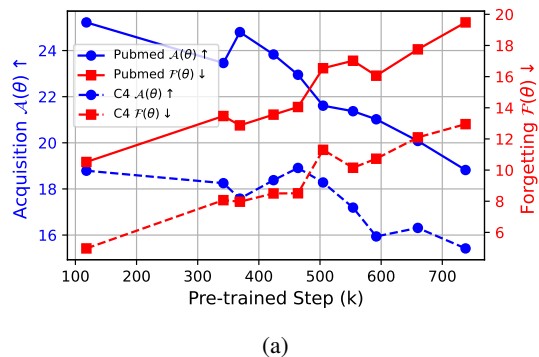

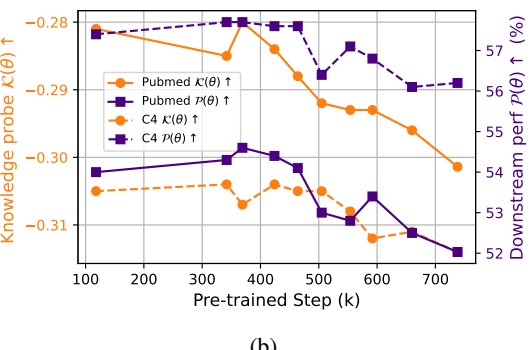

(a)

(b)

Figure 13: (a) Rates of knowledge acquisition $\mathcal{A}(\theta)$ and forgetting $\mathcal{F}(\theta)$ (b) End performance of knowledge probe $\mathcal{K}(\theta)$ and downstream task $\mathcal{P}(\theta)$ of Pubmed(solid line) and C4(dotted line) corpus across different model states. The x-axis represents the initial model state used for continual training.

## B.3 FREQUENCY OF KNOWLEDGE INJECTIONS

We divide the experiment into two settings: the *once* setting, where knowledge is injected a single time, and the *paraphrase* setting, where knowledge is injected ten times using ten paraphrased paragraphs. Figure 11 presents knowledge acquisition results based on the frequency of injections. Knowledge acquisition and final performance generally follow similar trends in both settings, with models in the later stages of pretraining showing the lowest performance. However, the performance and acquisition rate of *once* setting lag behind those of *paraphrase* setting. Also, notably, for models in the final stage of pretraining, the acquisition rate in the *once* setting was negative. This indicates that the log probability of the injected knowledge decreased, preventing successful incorporation of the new knowledge. In other words, even the knowledge injected during continual learning is subject to forgetting throughout the continual learning process.

### B.4 KNOWLEDGE ACQUISITION & FORGETTING

### B.4.1 BASELINE SETUP

Our base experiments are conducted using a hyperparameter configuration most closely aligned with continual knowledge learning studies, specifically with a batch size of 128, a learning rate of 4e-4, while training single-epoch of PubMed corpus. We use adamW optimizer ($\beta$ = 0.9, 0.95, weight decay= 0.1), cosine LR scheduler with warmup=0.05, and set maximum sequence length as 1024. We randomly selected 204,800 instances from the PubMed and C4 datasets and matched the sequence length to 1,024 tokens by concatenating instances. This resulted in a training dataset consisting of approximately 210 million tokens.

As analyzed in Section 4.2, final performance generally deteriorates as later-stage models are utilized as the initial model. Figure 12 illustrates the model's initial performance before continual learning, as well as its performance afterward. Models in the later stages of pretraining exhibit superior language modeling abilities before continual learning, as evidenced by the lower log probability for the newly injected knowledge (dotted line). However, after continual learning, their performance deteriorates compared to models from the earlier stages of pretraining (solid line). Similarly, the downstream task performance of the later-stage models was better initially, but after continual learning, their performance declined more than that of the earlier-stage models.

### B.4.2 VARIOUS SETTINGS

We observed consistently, even with altered hyperparameter settings, that models in the later stages of pretraining struggle to learn new knowledge and retain existing knowledge.

**Training Dataset for Continual Knowledge Learning**   To investigate how the type of *new* knowledge affects performance, we further conducted experiments using the C4 corpus, which has a distribution more similar to the pretraining corpus, Dolma, compared to PubMed. Figure 13a indicates that the gap in acquisition rate between later-stage models and initial-stage models is larger when the new knowledge distribution differs significantly from the pretraining corpus: models trained with PubMed (-6.4%p) exhibit a more pronounced gap compared to those trained with C4 (-3.4%p).

All models, regardless of their pretraining stage, tend to perform better when continually pretrained with PubMed compared to C4. We hypothesize that this is because PubMed's different distribution from the pretraining corpus encourages the model to learn more new knowledge, enhancing its ability to acquire new information. However, the rate of improvement varies by model state. Later-stage models tend to show similar performance regardless of the type of new knowledge, suggesting a limit in their learning capacity. In contrast, initial-stage models exhibit a stronger ability to acquire knowledge when trained on a corpus with a different distribution, such as PubMed, demonstrating their greater adaptability in learning new and diverse information.

**Model Size**   To test the universal deterioration of knowledge acquisition and retention capabilities, we also experimented with the OLMo 7B model Groeneveld et al. (2024). In Figure 4a, the 7B model exhibits a clear trend of diminishing knowledge acquisition $\mathcal{A}(\theta)$ capabilities as pretraining progresses. This decline is accompanied by an increase in forgetting $\mathcal{F}(\theta)$, indicating that the model struggles to retain previously learned information as new data is introduced.

**Batch Size**   In Table 1 line (b), when the batch size is large, the influence of individual data points on the model update decreases, resulting in a lower acquisition rate, while forgetting is less pronounced. In the later stages of training, however, if sufficient learning rate warmup is not provided, the model appears to collapse.

| | $\mathcal{A}(\theta) \uparrow$ | | | | $\mathcal{F}(\theta) \downarrow$ | | | |
|---|---|---|---|---|---|---|---|---|
| Pretraining Step of $\theta$ | 118k | 369k | 554k | 738k | 118k | 369k | 554k | 738k |
| (a) Baseline | 25.2 | 24.8 | 21.4 | 18.8 | 10.5 | 12.9 | 17.0 | 19.5 |
| (b) BS :128 → 2048 | 8.9 | 7.8 | 1.0 | -92.2 | 2.2 | 3.5 | 5.5 | 44.9 |
| (c) lr : 4e-4 → 1e-3 | 17.3 | 15.7 | 10.7 | 5.0 | 16.7 | 20.3 | 23.0 | 23.5 |
| (d) lr : 4e-4 → 1e-4 | 21.6 | 23.0 | 23.7 | 22.6 | 2.4 | 3.2 | 4.6 | 7.3 |
| (e) ep : 1 → 3 | 28.8 | 28.0 | 25.9 | 24.5 | 14.8 | 17.4 | 20.1 | 23.1 |

Table 1: Knowledge Acquisition $\mathcal{A}(\theta)$[%] and Forgetting $\mathcal{F}(\theta)$[%] across different Continual Learning hyperparameters.

| | $\mathcal{K}(\theta) \uparrow$ | | | | $\mathcal{P}(\theta) \uparrow$ | | | |
|---|---|---|---|---|---|---|---|---|
| Pretraining Step of $\theta$ | 118k | 369k | 554k | 738k | 118k | 369k | 554k | 738k |
| (a) Baseline | -0.281 | -0.280 | -0.293 | -0.301 | 54.0 | 54.6 | 52.8 | 52.0 |
| (b) BS :128 → 2048 | -0.342 | -0.344 | -0.368 | -0.714 | 59.0 | 60.5 | 60.1 | 35.6 |
| (c) lr : 4e-4 → 1e-3 | -0.310 | -0.314 | -0.332 | -0.353 | 50.3 | 49.9 | 49.0 | 49.4 |
| (d) lr : 4e-4 → 1e-4 | -0.294 | -0.287 | -0.284 | -0.287 | 58.9 | 60.7 | 60.6 | 59.9 |
| (e) ep : 1 → 3 | -0.267 | -0.268 | -0.276 | -0.280 | 51.4 | 51.7 | 50.8 | 49.7 |

Table 2: Probe Performance $\mathcal{K}(\theta)$ and average performance on six downstream task $\mathcal{P}(\theta)$ across different Continual Learning hyperparameters.

**Learning Rate**   In Table 1 line (c), when the learning rate is increased, not only is the acquisition rate lower but forgetting becomes more pronounced. Conversely, when the learning rate is very small (in line (d)), the models in the late stage show improved acquisition performance with the least gap compared with initial models, but still perform worse than the mid-stage model.

**Epoch**   In Table 1 line (e), as the number of epochs increases, leading to more repetitions, acquisition improves with the cost of more forgetting.

### B.5   TRAINING DYNAMICS OF PYTHIA

Figure 14 shows the training dynamics of Pythia Biderman et al. (2023) 1.4B and OLMo 1B. We observe that the overall training dynamics of Pythia and OLMo are similar. However, Pythia tends to conclude early in training, using only 10% of training tokens compared to OLMo. As a result, Pythia struggles to capture the full dynamics of the language model during pretraining. This is because models in the early stage of training are still influenced by a large portion of randomly initialized parameters, which leads to different behavior compared to models that have undergone more training steps and reached a stable state. For OLMo, based on our analysis, we assume the model reaches a stable point around 118k steps (15% of the training dataset). However, Pythia's final step occurs before this stable point, causing its trend to resemble that of OLMo but deviate from the trend observed in Section 4, where knowledge acquisition improves until the stable point and then decreases. We hypothesize that previous studies using intermediate checkpoints during

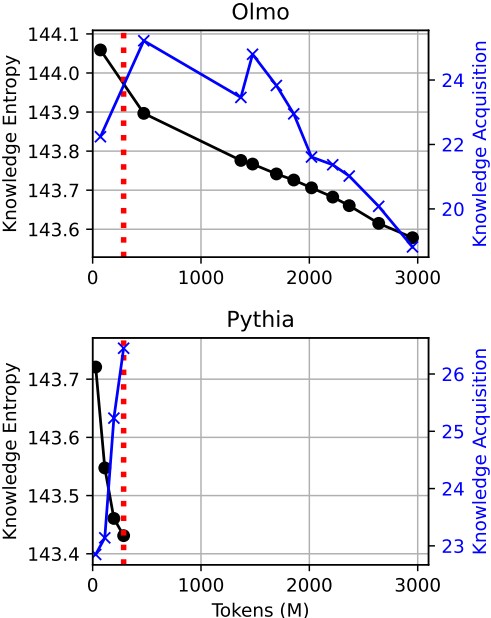

Figure 14: Knowledge Entropy and Rates of knowledge acquisition $\mathcal{A}(\theta)$ of OLMo 1B and Pythia 1.4B model across different pretraining stages

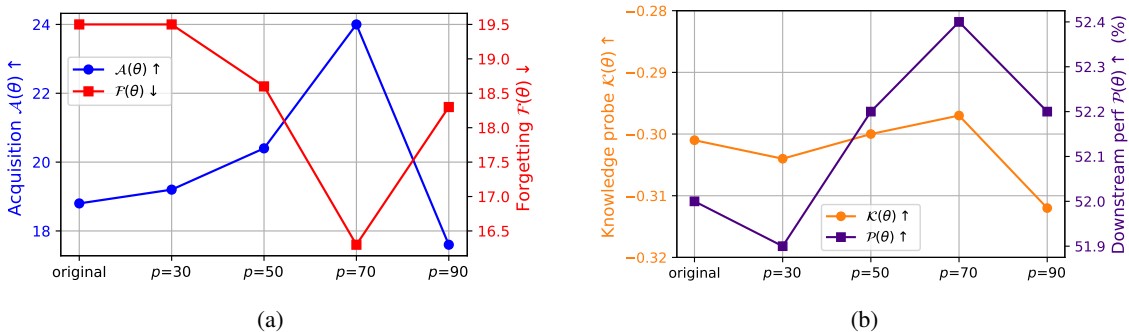

Figure 15: (a) Rates of knowledge acquisition $\mathcal{A}(\theta)$ and forgetting $\mathcal{F}(\theta)$ (b) End performance of knowledge probe $\mathcal{K}(\theta)$ and downstream task $\mathcal{P}(\theta)$ after training with varying $p$ (x-axis) while $q$ is fixed as 1.

pretraining (Chang et al., 2024; Sun & Dredze, 2024) also only measured with OLMo without Pythia for the same reason.

## B.6    RESUSCITATION EXPERIMENT: VARYING $p$ WHILE FIXING $q$

Figure 15 shows the overall performance when $q$ was fixed at 1, meaning that the lowest $p\%$ of coefficients were scaled to converge toward the layer's mean. It is shown that increasing $p$ to activate a larger portion of

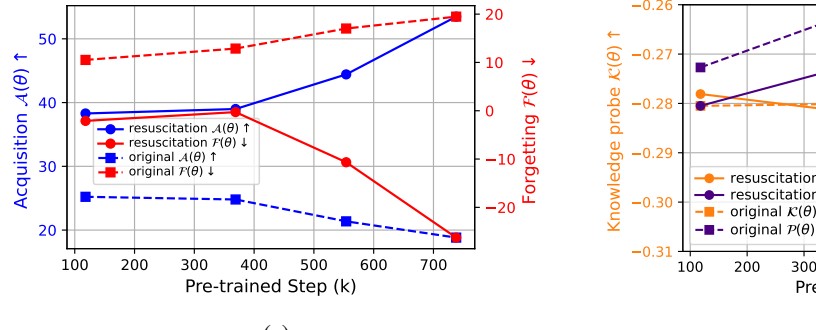

Figure 16: (a) Rates of knowledge acquisition $\mathcal{A}(\theta)$ and forgetting $\mathcal{F}(\theta)$ (b) End performance of knowledge probe $\mathcal{K}(\theta)$ and downstream task $\mathcal{P}(\theta)$ of original continual learning (dotted line) and resuscitation (solid line) method where $p$ is fixed as 50 and $q$ as 2, across different model states. The x-axis represents the initial model state used for continual training.

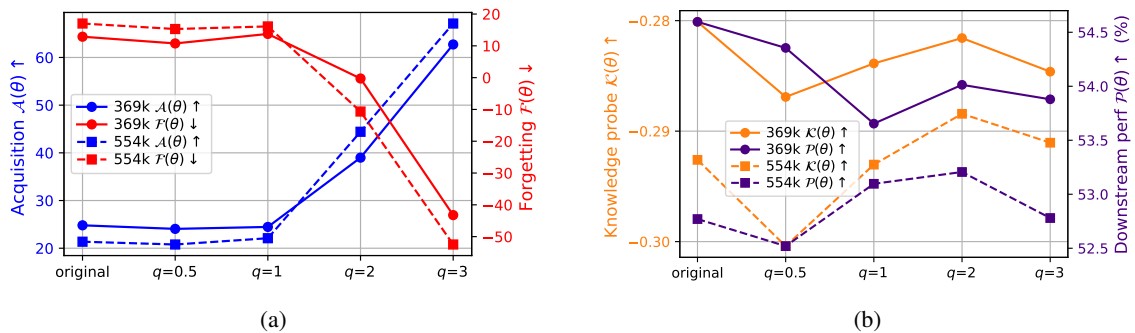

Figure 17: (a) Rates of knowledge acquisition $\mathcal{A}(\theta)$ and forgetting $\mathcal{F}(\theta)$ (b) End performance of knowledge probe $\mathcal{K}(\theta)$ and downstream task $\mathcal{P}(\theta)$ of model from 369k steps (solid line) in pretraining schedule and 554k (dotted line) when varying the amplifying factor $q$ (x-axis) while $p$ is fixed as 50.

inactive parameters generally led to improved performance. However, interestingly, setting $p$ too high(e.g., at 90) negatively impacted performance, likely due to the unintended effect of reducing the parameters in already active regions.

### B.7    RESUSCITATION EXPERIMENT ACROSS PRETRAINING STEPS

Figure 16a illustrates the overall performance when $q$ was fixed to 2 and $p$ to 50, across different pretraining stages of the original model. The effect of the resuscitation method becomes more pronounced as the original model progresses to later stages of pretraining, as indicated by the transition from the dotted line to the solid line. We hypothesize that this is because late-stage models tend to rely on a smaller subset of memory sources and thus benefit from a broader scope of activation enabled by the resuscitation method. As shown in Figure 16b, end performance deteriorates when the beginning model is initial (118k) and mid (369k) stage model, indicating that resuscitation may impair performance when the model's knowledge entropy is not sufficiently low. This trend of the resuscitation showing a more positive effect for models in the later stage of pretraining can also be seen in Figure 17, which shows the results when varying $q$ while fixing $p$ at 50:

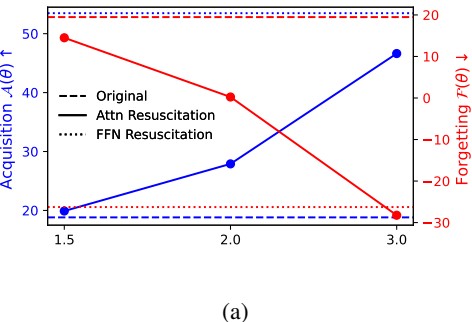 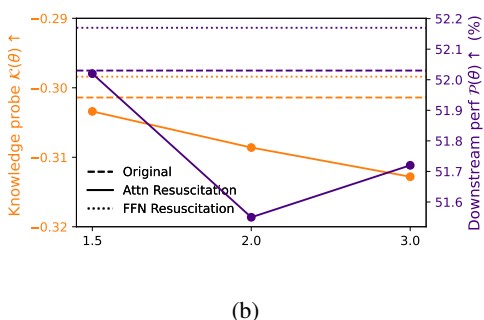

(a)                          (b)

Figure 18: (a) Rates of knowledge acquisition $\mathcal{A}(\theta)$ and forgetting $\mathcal{F}(\theta)$ (b) End performance of knowledge probe $\mathcal{K}(\theta)$ and downstream task $\mathcal{P}(\theta)$ when resuscitation is applied at **attention layers**, with the x-axis representing temperature. The dotted line indicates performance when resuscitation is applied to the feed-forward layer with $p$=50, $q$=2. The dashed line represents the original performance without any resuscitation.

performance deteriorates when running continual learning on the model from 369k, while improvement of performance with larger $q$ is observed when the model is from 554k.

## B.8    RESUSCITATION EXPERIMENT ACROSS ATTENTION LAYERS

In Figure 3, we observe that as pretraining progresses, not only knowledge entropy but also attention entropy decreases. We explore whether extending the resuscitation method to the attention layers could similarly improve knowledge acquisition and retention. For the feed-forward layer, we can adjust knowledge entropy by tuning the parameters at positions with low activation values. However, since attention entropy reflects the probability distribution over token positions, the same resuscitation method cannot be directly applied to the attention layers. To artificially increase attention entropy, we employed temperature scaling on the softmax function in the attention calculation, which reduces the sparsity of the probability distribution.

We experimented with temperature values ranging from 1.5 to 3.0. Figure 18 shows that resuscitation in the feed-forward layer consistently results in the highest knowledge acquisition rate and generally leads to the best knowledge retention. This trend is observed in both knowledge probe results and downstream performance. These findings suggest that knowledge entropy plays a critical role in influencing both knowledge acquisition and forgetting, thereby driving the observed differences in performance.

