# OpenReview forum: "Knowledge Entropy Decay during Language Model Pretraining Hinders New Knowledge Acquisition"
_ICLR.cc/2025/Conference — ICLR 2025 Oral_

### Official Review · Reviewer_yokT · 2024-10-17

**Soundness:** 3
**Presentation:** 3
**Contribution:** 3
**Rating:** 8
**Confidence:** 4

**Summary:**

The paper introduces knowledge entropy as a novel concept for the quantification of model representation across Large Language Models. The authors show that as the training of a large model progresses, the knowledge tends to concentrate in a smaller set of "memory vectors", making the process of knowledge acquisition difficult (decrease in plasticity), and the model more prone to forgetting prior knowledge as memory vectors are forced to be constantly rewritten. Experimental results show that it is indeed the case that models in the later stages of their training are more resilient to learning new knowledge, and tend to easily forget the knowledge that they already have. In addition, by resuscitating the inactive vectors, the authors force a later-stage model to ameliorate both the aforementioned concerns in the continual learning process.

Overall, I think this is a good paper with interesting results that can have further applications for the community. Some concerns and questions still remain as outlined below, but I am willing to increase my score once they are addressed.

**Strengths:**

- The proposed framework of categorizing knowledge retention in terms of the model's knowledge entropy is interesting albeit somewhat expected as it is generally understood that moving away from established weight matrices becomes more difficult as the training progresses.
- The concept of entropy in this case is both well-defined and intuitive, essentially showing how variant the coefficients are with respect to each input token given an instance.
- The experimental results confirm the theoretical hypothesis, strongly indicating that knowledge entropy is associated with both learning and forgetting.

**Weaknesses:**

- The usage of the term "entropy" in this case can be confusing in my opinion. Usually, this term implies some form of "surprise" with respect to information. However, in this case, it seems to refer to the idea of source broadness.
- I think this is a paper that could benefit greatly from further evaluation with respect to knowledge entropy. Although it is expected that the same behavior is observed across models, I would like to know if different architectures affect the observed behavior and if different training time design choices can regulate the knowledge entropy on a case-by-case basis.
- Further discussion is required regarding the correspondence of knowledge entropy and forgetting. Rather than being a direct consequence of knowledge entropy, can it be the case that the reason behind sharper declines in knowledge retention is due to the model's richer knowledge representation between concepts at later stages of the training and thus being more prone to knowledge loss when small changes are made? More specifically, forgetting at a higher rate at later stages can be expected simply due to having more knowledge encoded in comparison to having very little knowledge in earlier stages.
- Minor grammatical errors that can be fixed in the camera-ready version.
- I believe that Algorithm 1 can be rewritten to better relay the information contained. Namely, by giving a better explanation of each operation (especially with respect to $t$).

**Questions:**

- Given that consistent behavior is observed across all layers, do you think we can use only a subset of layers to determine the model's proneness to learning new knowledge? For example, only focusing on layers that mainly encode semantic knowledge as shown in the literature.
- Related to the question above, is the same trend observed across different forms of knowledge and is there a difference between various knowledge forms (syntactic knowledge versus semantic knowledge for example)?
- Given the minor negative effects of resuscitation of inactive vectors, do you think this is a viable method for adaptation in continual learning frameworks in generals to minimize forgetting and increasing knowledge acquisition? More specifically, what negative effects can this artificial resuscitation have?
- In Figure 5(b), what is the reason behind the decrease in the downstream task performance while a drastic decrease in forgetting is also observed? Shouldn't a low forgetting rate lead to better general-purpose performance?

---

> ### Author Response · Authors · 2024-11-20
> **Response to Reviewer yokT**
>
> Dear Reviewer yokT,
>
> Thank you for your valuable time in reviewing our paper. We appreciate recognizing the novelty of our work and experimental results.
>
> >The usage of the term "entropy" in this case can be confusing in my opinion. Usually, this term implies some form of "surprise" with respect to information. However, in this case, it seems to refer to the idea of source broadness.
>
> The reviewer raises a question about whether the observed behavior regarding the knowledge entropy holds across different models, architectures, or training time design choices. To address this, we further test the behavior of knowledge entropy using the Pythia model (1.4b) with Dolma, we observed that the behavior remains consistent. We add the result of Pythia knowledge entropy and update the paper where the figure is in Figure 6 of the revised version of the paper. We kindly ask the reviewer to check the figure of the trend from the updated paper as we cannot attach figures in the rebuttal. Also in the original and the revised paper, to investigate whether the findings hold for various cases, we conducted experiments where we changed the activation function from SwiGLU to ReLU (Figure 9) and adjusted model sizes (1b, 7b) in Figure 2.
>
> Due to computational constraints and the lack of models that provide intermediate checkpoints, we were unable to explore all possible training time design choices. However, based on the results from the various combinations we tested, we expect that the general trend would persist across different training scenarios.
>
> >Further discussion is required regarding the correspondence of knowledge entropy and forgetting. Rather than being a direct consequence of knowledge entropy, can it be the case that the reason behind sharper declines in knowledge retention is due to the model's richer knowledge representation between concepts at later stages of the training and thus being more prone to knowledge loss when small changes are made? More specifically, forgetting at a higher rate at later stages can be expected simply due to having more knowledge encoded in comparison to having very little knowledge in earlier stages.
>
> The reviewer suggests that the decline in knowledge retention may be due to the model’s richer knowledge representation at later stages of training since the later models are likely to contain more knowledge compared to initial models. We agree that models in later stages of training would typically encode more knowledge than earlier stages. However, we believe the key factor is not just the amount of knowledge, but how it is stored and utilized.
>
> Our resuscitation experiment highlights this distinction. By altering the knowledge entropy while keeping the same parameter state (the final checkpoint), we observed significant differences in knowledge retention. This suggests that how knowledge is stored and accessed plays a crucial role. Specifically, when the model relies on a small set of certain memory vectors, the rate of forgetting is higher.
>
> >Minor grammatical errors that can be fixed in the camera-ready version.
>
> Thank you for pointing this out. We will address the minor grammatical errors in the camera-ready version.
>
> >I believe that Algorithm 1 can be rewritten to better relay the information contained. Namely, by giving a better explanation of each operation (especially with respect to t).
>
> Thank you for the feedback. We will revise Algorithm 1 to make it clearer and more explainable, particularly with respect to t, as the reviewer suggested.
>
> >Given that consistent behavior is observed across all layers, do you think we can use only a subset of layers to determine the model's proneness to learning new knowledge? For example, only focusing on layers that mainly encode semantic knowledge as shown in the literature.
>
> The reviewer inquires if the behavior of a subset of layers can determine the model’s ability to learn and retain knowledge, especially in terms of the critical layers that have been proposed as encoding semantic knowledge.
>
> According to Meng et al. (2022)[1], feed-forward layers in the middle range of a model play a crucial role in processing the last token of the subject name for factual mediation. Meng et al. (2023)[2] and Meng et al. (2022) identified this range as layers 4 to 9 in GPT-J (6B), which has 28 layers, and up to layer 15 in GPT-2 XL (1.5B), which has 48 layers.
>
> As shown in Figure 8 (Appendix A.2 in the revised version of paper), the layers identified as critical for processing factual knowledge (3rd to 8th layers) exhibit the highest values of Knowledge Entropy and experience less reduction in knowledge entropy during pretraining. The overall reduction in knowledge entropy was driven by the other layers. Based on this observation, it seems that relying on only a subset of layers may not be representative enough to assess the model's overall ability.

---

> > ### Author Response · Authors · 2024-11-20
> >
> > >Related to the question above, is the same trend observed across different forms of knowledge and is there a difference between various knowledge forms (syntactic knowledge versus semantic knowledge for example)?
> >
> > The reviewer raises a question about whether the trend of Knowledge Entropy is consistent across various knowledge types, such as syntactic and semantic knowledge. We did additional analysis over the trend of knowledge entropy across various corpus sets and divided a single corpus in token level into syntactic and semantic tokens to observe the overall trend of Knowledge Entropy.
> >
> > While the Dolma dataset (training dataset) provides labeled data sources, downloading the entire dataset was infeasible due to storage and time constraint. Instead, we used the sampled RedPajama corpus (RedPajama-Data-1T-Sample[3]), which includes part of domains found in Dolma but provides smaller random samples. When measuring knowledge entropy for each corpus in RedPajama (arxiv, books, c4, cc, github, stackexachange, and wikipedia), we observe that overall trend tend to be similar: knowledge entropy tends to decrease as pretraining progresses. However, for some datasets, we observed instability in the knowledge entropy trend at mid-checkpoints.
> >
> > We hypothesize that this instability is related to the initial knowledge entropy values during pretraining. Specifically, datasets with lower initial knowledge entropy tend to exhibit more unstable trends compared to those with higher initial knowledge entropy. Nonetheless, for all datasets, knowledge entropy overall decreases when comparing the initial and final stages of pretraining. As the table with the result tend to be large and it is unable to include figures directly in the rebuttal, we have provided the figure at the following link (https://anonymous.4open.science/r/ke_temp-B4A6/ke_corpus.pdf). Sorry for the inconvenience and kindly ask the reviewer to review the figures from the link.
> >
> > This trend was also evident in our token-wise analysis. In Geva et al.(2022)[4], NLP experts annotated the role of value vectors based on the patterns of top-scoring tokens, classifying them into semantic and syntactic concepts. As the paper did not release the dataset and due to time and cost constraints, we utilized GPT-4 to classify tokens in 5,000 instances from the Dolma corpus into three categories: syntactic tokens, named entity tokens, and semantic tokens. Notably, many of the tokens labeled as named entities were pronouns rather than proper named entities.
> >
> > The results showed that Knowledge Entropy consistently decreases for all types of tokens. The figure containing the results is at https://anonymous.4open.science/r/ke_temp-B4A6/ke_token.pdf. It is noteworthy that syntactic tokens and named entity tokens demonstrated lower Knowledge Entropy. While the classification may not be perfectly precise, we conjecture that tokens that the model can predict with higher confidence tend to lead to a more decisive selection of memory vectors, leading to lower values of knowledge entropy.
> >
> > >Given the minor negative effects of resuscitation of inactive vectors, do you think this is a viable method for adaptation in continual learning frameworks in generals to minimize forgetting and increasing knowledge acquisition? More specifically, what negative effects can this artificial resuscitation have?
> >
> > The reviewer raises a question about what negative effects can exist due to our artificial resuscitation and whether it is a viable method for general cases.
> >
> > While our resuscitation method reduces the forgetting rate and improves knowledge acquisition, we observed that simply manipulating the model parameters at the initial state using Algorithm 1 can lead to significant performance degradation. Specifically, even though we achieve higher knowledge acquisition and lower forgetting, the performance tends to be lower compared to using a middle-stage model. This middle-stage model has higher knowledge entropy than the final-stage model, while having similar entropy levels to the resuscitated model.
> >
> > We hypothesize that this difference arises from the timing and nature of the parameter manipulation—whether it increases knowledge entropy or occurs during pretraining. Based on this, we recommend using the middle-stage model when new knowledge acquisition is required. However, we believe that refining the parameter manipulation beyond the basic approach in Algorithm 1 could reduce the initial performance degradation, leading to better overall performance, especially with larger q values.

---

> > > ### Author Response · Authors · 2024-11-20
> > >
> > > >In Figure 5(b), what is the reason behind the decrease in the downstream task performance while a drastic decrease in forgetting is also observed? Shouldn't a low forgetting rate lead to better general-purpose performance?
> > >
> > > The reviewer raises a question about the reason behind the decrease in downstream task performance while decrease in forgetting rate in Figure 5(a). Figure 5(a) shows the improvement and degradation rate of knowledge acquisition and forgetting, respectively, which represent the *difference relative to the initial state* (equation 4, 5 in Appendix B.2 in the revised version of the paper). Importantly, the initial performance varies with q due to the scaling effect introduced in Algorithm 1. This means that even with lower forgetting rates, downstream performance may not always increase for larger q values because the initial performance for different q values is not the same.
> > >
> > > In Figure 5(b), we observed that for larger q values, the higher acquisition rate and lower forgetting rate generally led to better knowledge probing and downstream performance compared to the original state. For the case of q=3, as the parameter manipulation becomes more severe with larger q leading to a larger initial performance drop, we could see that even with a low forgetting rate, it resulted in lower downstream performance than the original.
> > >
> > > We would like to emphasize that the main goal of the resuscitation experiment was to investigate whether changes in knowledge entropy drive the differences in knowledge acquisition and forgetting rates, while keeping the same parameter state and altering knowledge entropy using Algorithm 1. As shown in Figure 5(a), increasing the q value leads to higher knowledge acquisition and forgetting rates, supporting our hypothesis.
> > >
> > > **References**
> > >
> > > [1] Locating and Editing Factual Associations in GPT, Meng et al.,
> > >
> > > [2] Mass Editing Memory in a Transformer, Meng et al.,
> > >
> > > [3] https://huggingface.co/datasets/togethercomputer/RedPajama-Data-1T-Sample
> > >
> > > [4] Transformer Feed-Forward Layers Build Predictions by Promoting Concepts in the Vocabulary Space, Geva et al.,

---

> > > > ### Comment · Reviewer_yokT · 2024-11-23
> > > > **Reviewer Response**
> > > >
> > > > Thank you for the thorough response to my concerns. Given that the phenomenon seems to hold across different test dimensions, as well as its possible consequences for the general interpretability community, I would like to recommend this paper be accepted. However, I am still slightly unconvinced regarding the causal relation between knowledge entropy and knowledge acquisition/forgetting in Large Language Models, requiring future work to fully confirm/deny this via further evaluation and maybe a more theoretical approach. I additionally believe that future work on the relationship between knowledge entropy and different knowledge dimensions would hold merit.
> > > >
> > > > I suggest discussing the findings with respect to knowledge types either briefly in the main paper or in the appendix.

---

> > > > > ### Author Response · Authors · 2024-11-23
> > > > >
> > > > > The authors sincerely thank you for your positive feedback on the topic we have discussed. We also believe that a more thorough exploration of the relationship between knowledge entropy and different knowledge dimensions could be pursued as future work. We will include our additional findings in the camera-ready version as you suggested.

---

### Official Review · Reviewer_EdjN · 2024-10-29

**Soundness:** 3
**Presentation:** 2
**Contribution:** 3
**Rating:** 8
**Confidence:** 3

**Summary:**

This paper investigates how knowledge acquisition and forgetting evolves through an LLM's pretraining. Building on the work of Chang et al 2024 and Geva et al 2021 and 2022, the authors introduce a concept of knowledge entropy and show that low entropy, which corresponds to poor knowledge acquisition and high forgetting rate, occurs in late stages of pretraining while high entropy occurs for initial stages of pretraining.

**Strengths:**

This work exposes an important feature of pretraining and contributes valuable insights to LLM pretraining.
While the definition of knowledge entropy to this reviewer is not completely clear, it is an important feature of it that it patterns with other notions of entropy in models like attention entropy or next token prediction entropy.   The work on knowledge acquisition and forgetting is potentially very important for how we view model training.


The empirical results are solid and convincing to this reviewer.

**Weaknesses:**

This paper has the potential to make substantial contributions to the NLP and ML communities.  But I believe it still could benefit from further work.

The talk of broad and particular memory vectors in the introduction is a bit unclear.

In calculating entropy, the authors assume the feed forward network has the form
FFN(x) = f(x · K^T ) · V
but this leaves out important bias terms that might perhaps affect the calculations.  Can the authors say more about this?


There are some important connections with Chang et al 2024 that need to be brought out more.  As it stands, it seems as though much of the heavy lifting is done in Chang et al, and this paper provides an incremental improvement to that paper.  There needs to be a discussion of how this paper goes beyond the previous one.

Also it is important to not oversell the information contained in the so called memory vectors.  In particular, given the experiments with human interpreters given the strings that affect these memory vectors the most,  what we get are suggestive interpretations of what's stored, that is suggestive to those particular human interpreters.  As far as I know Chang et al shows mostly  it's not clear that the model can make use of them in the way they are interpreted.

While it's pretty clear intuitively what attention entropy or next token prediction entropy are, it's less clear at least to this reviewer what the intuition is behind defining entropy in terms of the coefficients of the outputs of a given FFN layer in the model.   How does the knowledge entropy so defined link up with the suggestion that "models in the later stage of pretraining tends to engage with a narrower
range of memories, relying more heavily on specific memory vectors rather than accessing and integrating
knowledge from a broader range of memories"?

Another element that needs a bit more work is the discussion of knowledge acquisition and forgetting.  This relies on work from Chang et al. but to be more self-contained the authors could explain in a bit more detail what the probes are doing (and why, for instance, 15 different probes?), or rather summarize Chang et al's contributions more clearly with respect to the contributions of the present work.

That part as it stands leaves this reader wanting more information.  For instance, what is the target span for the probes?  Also is \theta_{pt} unique? Is there only one such check point and what about \theta_{cl}?  Is \theta_{cl} when the continual learning stops, if it does, or what?  Is  \theta_{cl} then the model after pre training?   There is more material in the appendix on this but it would help the reader's understanding of this important concept of knowledge acquisition and knowledge loss  to move some of that discussion into the body of the paper.

Some minor stylistic corrections that will render the paper more readable:

Works ---> work   (`work' is still a mass known in English, as far as I know)
coefficient of j-th token  ---> coefficient of the  j-th token

into probability distribution --> into a probability distribution

We also experiment that the trend persists ---> We also show experimentally that the trend persists

we inject FICTIONAL KNOWLEDGE dataset  ---> we inject elements of the FICTIONAL KNOWLEDGE dataset??

of target span  ---> of the target span

Equation and detailed explanation is
presented  ---> please fix the English

Hoyeon Chang, Jinho Park, Seonghyeon Ye, Sohee Yang, Youngkyung Seo, Du-Seong Chang, and Minjoon
Seo. How do large language models acquire factual knowledge during pretraining? arXiv preprint
arXiv:2406.11813, 2024.

**Questions:**

Please address the questions about bias, and also about how knowledge entropy links to intuitions about types of memories.  Finally please answer the questions about probes and knowledge acquisition.
Could you more clearly articulate how this paper builds on chang et al with respect to the part on knoweldge acquisition and forgetting?

---

> ### Author Response · Authors · 2024-11-20
>
> Dear reviewer EdjN,
>
> Thank you sincerely for taking the time to share your valuable insights. We appreciate your recognition of the importance of our work and its potential to contribute significantly to the understanding of model training.  Your acknowledgment of the solid empirical results and the potential for substantial contributions to the community is truly encouraging. The authors will incorporate your comments to improve the quality of the paper.
>
> >The talk of broad and particular memory vectors in the introduction is a bit unclear.
>
> The reviewer highlighted the ambiguity in the concept of broad and particular memory vectors mentioned in the introduction. Our intention was to introduce the idea that a model's processing can be understood as integrating parametric memories, which we later define as memory vectors(the row vectors of the final matrix in feed-forward layers), building on prior literature that interprets feed-forward layers as key-value memories ([1] Geva et al., 2021). By broad and particular memory vectors, we were referring to the model's behavior of whether the memory vectors are combined in uniform distribution (most memory vectors are used with low coefficient) or skewed distribution (only part of the memory vectors are used with high coefficient), respectively. We will clarify the definition of these terms in the final version to ensure better understanding.
>
> >In calculating entropy, the authors assume the feed forward network has the form FFN(x) = f(x · K^T ) · V but this leaves out important bias terms that might perhaps affect the calculations. Can the authors say more about this?
>
> The reviewer raises a question regarding the exclusion of the bias term in our entropy calculation; how would the calculation change with inclusion of bias term.
> We would like to clarify that the model we used in our experiments, OLMo, does not include a bias term. As noted in the OLMo paper[2]: “No biases. Following LLaMA, PaLM, and others, we exclude all bias terms from our architecture in order to improve training stability.”
>
> We believe that even with a bias term, the intuition and interpretation regarding knowledge entropy remain unchanged. Specifically, the output of $f(\mathbf{x} \cdot K^T)$ represents the weights for the linear combination of the corresponding vectors in the $V$ matrix and knowledge entropy measures the distribution of these weights. With a bias term, a memory vector always having a coefficient as 1 is added, which should not affect the distribution of the other coefficients. Thereby, calculation of knowledge entropy does not take into account the bias vector.

---

> > ### Author Response · Authors · 2024-11-20
> >
> > >There are some important connections with Chang et al 2024 that need to be brought out more. As it stands, it seems as though much of the heavy lifting is done in Chang et al, and this paper provides an incremental improvement to that paper. There needs to be a discussion of how this paper goes beyond the previous one.
> >
> > The reviewer raises a concern that the difference between our work and Chang et al., 2024 [3] needs further clarification. While both studies investigate the knowledge acquisition behavior during LLM training, their motivations and focus diverge, leading to distinct experimental setups.
> > Chang et al.2024 primarily analyzes patterns of knowledge acquisition specifically *during the pretraining process*, addressing the question of “how” language models acquire knowledge during pretraining. In contrast, our work seeks to uncover the *factors* that make it increasingly difficult for LLMs to acquire new knowledge as pretraining progresses, tackling the question of “why” later-stage models exhibit difficulty in learning new knowledge.
> > Although Chang et al., 2024 also observe that later-stage models do not show better performance in acquiring new knowledge compared to earlier stages, they do not explore the underlying reasons for this trend. Our study aims to fill this gap, providing insights into the factors driving this phenomenon.
> > Due to the difference in the motivation, while we directly adopt the dataset, knowledge injection, and measurement methodology from Chang et al. (2024), there are several differences in the experimental setup, which are summarized as the table below.
> >
> > | **Aspect**                | **Chang et al. (2024)**                        | **Ours**                                             |
> > |---------------------------|-----------------------------------------------|-----------------------------------------------------|
> > | **Training mode at question** | Pretraining                                 | Continual Knowledge Learning; adaptation to new corpus |
> > | **Training corpus**       | Dolma                                       | Pubmed, C4                                          |
> > | **Optimizer / Learning rate** | Loaded from original schedule              | Newly initialized, identical for all checkpoints for comparison |
> > | **Checkpoints employed**  | 3 models trained for 40k, 118k, 358k steps   | 10 models trained at intervals ranging from 118k to 738k steps |
> >
> > We have updated the paper including more detailed explanation of the difference between Chang et al., 2024 and our work in the related works section in revised version of our paper.
> >
> > >Also it is important to not oversell the information contained in the so called memory vectors. In particular, given the experiments with human interpreters given the strings that affect these memory vectors the most, what we get are suggestive interpretations of what's stored, that is suggestive to those particular human interpreters. As far as I know Chang et al shows mostly it's not clear that the model can make use of them in the way they are interpreted.
> >
> > The reviewer raises concern about the interpretation of the information stored in memory vectors and the importance of not overstating our findings.
> > To clarify, we do not claim that specific memory vectors encode specific information or are selectively activated in particular contexts. Instead, our analysis highlights the observed tendency of certain memory vectors to exhibit higher average activation across the overall corpus. This behavior is a characteristic of the language model and is not directly related to the specific information stored in the memory vectors. Our focus is on understanding how this tendency, viewed from the perspective of concentrated parameter updates during model training, affects knowledge acquisition and retention.

---

> > > ### Comment · Reviewer_EdjN · 2024-11-23
> > >
> > > OK thanks for your answer; it helps me understand better how your paper is positioned..  I think it would be helpful to really emphasize the 'why' question that you're answering in contrast to Chang et al.

---

> > > > ### Comment · Reviewer_EdjN · 2024-11-26
> > > >
> > > > In thinking about your reply a bit further, I don't really see a causal explanation in what you did but rather a reinterpretation of what's going on.  The latter is weaker than a causal explanation but can still be useful.  Would you agree?

---

> > > > > ### Author Response · Authors · 2024-11-26
> > > > > **Response to Reviewer EdjN**
> > > > >
> > > > > Dear reviewer EdjN,
> > > > >
> > > > > Thank you for your additional inquiry about the causal relation between decreasing knowledge entropy and capability to acquire and retain knowledge. We acknowledge that the original submission lacked the explanation of the relationship, so we included the intuition behind it in our revised paper in Appendix A.1.
> > > > >
> > > > > Our analysis is based on the intuition that the coefficients in the linear combination of memory vectors determine how the corresponding memory vectors are updated.
> > > > > As defined in Equation 1, the FFN output ($FFN(\mathbf{x}) = f(\mathbf{x} \cdot K^T) \cdot V$)  is a linear combination of memory vectors $\mathbf{v}_{i=1,\cdots,m} \in \mathbb{R}^{d}$
> > > > >
> > > > > (the row vectors of $V \in \mathbb{R}^{m \times d}$ ),
> > > > > where the coefficients $c_i$ are given by $f(\mathbf{x} \cdot K^T)$.
> > > > > In other words, $FFN(x)=c_1\mathbf{v_1}+c_2\mathbf{v_2}+\cdots+c_m\mathbf{v_m}$. Within a given layer, since the operations beyond the FFN layer and the input to the FFN layer remain consistent across all memory vectors, the coefficients act as scaling factors for the gradient by the chain rule. During training, the gradient $\frac{ \partial L}{\partial \mathbf{v_{i,j}}}$ for $ i = 1, 2, \ldots, m$ and $ j = 1, 2, \ldots, d$ can be decomposed as: $$\frac{ \partial L}{\partial \mathbf{v_{i,j}}} = \frac{ \partial L}{\partial FFN(x)}\cdot \frac{ \partial FFN(x)}{\partial v_{i,j}}$$
> > > > >
> > > > > Here, $\frac{ \partial L}{\partial FFN(x)}$ is the same for all $\mathbf{v_i}$, meaning that the relative magnitude of the gradient depends on $\frac{ \partial FFN(x)}{\partial v_{i,j}} = c_i$. Thus, larger coefficients result in proportionally larger gradients being applied to the corresponding memory vectors, amplifying their updates during backpropagation.
> > > > >
> > > > > As pretraining progresses, a spikier coefficient distribution—captured by decreasing knowledge entropy—implies that gradient updates become increasingly concentrated on specific positions where the average coefficients are larger. This centralization can affect the model's ability to evenly utilize its memory capacity, impacting knowledge acquisition and retention.
> > > > >
> > > > > We hope this comment could address your question and are happy to discuss more if any remaining concerns. We sincerely appreciate your valuable feedback.

---

> > > > > > ### Comment · Reviewer_EdjN · 2024-11-27
> > > > > >
> > > > > > OK this is very helpful.  I think if you can clear up your connections with Chang et al and detail this explanation idea better in the paper I'm for it.

---

> > > > > > > ### Author Response · Authors · 2024-11-27
> > > > > > >
> > > > > > > We sincerely appreciate your encouraging comment. We will make sure to include the connections with Chang et al and the explanation in more detail in the camera-ready version. Thank you again for your valuable feedbacks.

---

> ### Author Response · Authors · 2024-11-20
>
> >While it's pretty clear intuitively what attention entropy or next token prediction entropy are, it's less clear at least to this reviewer what the intuition is behind defining entropy in terms of the coefficients of the outputs of a given FFN layer in the model. How does the knowledge entropy so defined link up with the suggestion that "models in the later stage of pretraining tends to engage with a narrower range of memories, relying more heavily on specific memory vectors rather than accessing and integrating knowledge from a broader range of memories"?
>
> The reviewer raises a question about the intuition behind defining entropy in terms of coefficients and how this definition connects to the observed behavior of the model.
> Building on top of previous research(Geva et al. (2021) [1]) viewing FFN layer as key-value memory, we wondered how the behavior of selecting values evolve as the model attains more parameteric knowledge during pretraining. Specifically, if more knowledge is stored in the last matrix of FFN layer as pretraining progresses, the model trained with more tokens will select relevant memories with more certainty. Thus we proposed knowledge entropy to measure this certainty, spiky distribution of “keys”.
>
> In the paper, as shown in Figure 1, we referred the memory vectors to the last projection matrix of FFN layer and the memory coefficients are input to the last projection. Knowledge entropy measures the sparsity of the coefficients in an FFN layer; while higher knowlege entropy implies uniform (less certain) distribution of the coefficients, integrating broader range of memory vectors, lower knowlege entropy captures spiky(more certain) distribution, focusing on certain range of memory vectors.
>
> The mechanism behind correlation of Knowledge Entropy and capability to acquire and retain knowledge is that the gradient of the loss is propagated to the memory vectors in the FFN layer in proportion to these coefficients. As the coefficients become more concentrated on specific positions, they may repeatedly overwrite the highly activated regions of the parameters rather than distributing across all parameters, thereby limiting the model’s overall knowledge capacity. Our findings reveal a consistent decrease in Knowledge Entropy as pretraining progresses, indicating that the model increasingly relies on certain memory vectors. When these late-stage model checkpoints are trained continually, the overall performance of knowledge acquisition and retention can be constrained due to the limited capacity. We included the intuition behind the definition of Knowledge Entropy and its impact on training in the Appendix A.1 in the revised version of the paper.
>
> >Another element that needs a bit more work is the discussion of knowledge acquisition and forgetting. This relies on work from Chang et al. but to be more self-contained the authors could explain in a bit more detail what the probes are doing (and why, for instance, 15 different probes?), or rather summarize Chang et al's contributions more clearly with respect to the contributions of the present work.
>
> The reviewer raises a question about the distinct contribution of our work from Chang et al [2]. Chang et al established a framework to measure LLM’s knowledge acquisition, and as we are measuring the model’s knowledge acquisition, we followed their dataset and evaluation metric. But please note that there is a difference in the training setups due to distinct motivations: while Chang et al analyzes how LLMs acquire knowledge during pretraining, we are rather interested in why LLMs lose this ability during pretraining.
>
> Here, we will briefly explain the knowledge injection and knowledge acquisition measurement framework established by Chang et al. The Fictional Knowledge dataset consists of two components: a fictional corpus, which contains new knowledge presented in a Wikipedia-style format, and corresponding probes that are cloze-task-style sentences to query the information within the corpus. The final span of each probe, referred to as the target span, is used to evaluate the model’s prediction probability, which serves as a measure of knowledge acquisition performance.
>
> In Chang et al., the probes are divided into three levels of difficulty, with five sentences created for each level. This results in 15 probes per corpus. The difficulty levels are as follows: 1) Memorization probes directly ask about sentences explicitly present in the fictional corpus. 2) Semantic generalization probes are paraphrased versions of the memorization probes to test the model’s understanding of meaning beyond surface forms. 3) Compositional generalization probes are designed to assess whether the model can integrate multiple pieces of knowledge from the fictional corpus.
> We have revised the detailed explanation of the evaluation dataset in Appendix B.1.

---

> ### Author Response · Authors · 2024-11-20
>
> >That part as it stands leaves this reader wanting more information. For instance, what is the target span for the probes? Also is \theta_{pt} unique? Is there only one such check point and what about \theta_{cl}? Is \theta_{cl} when the continual learning stops, if it does, or what? Is \theta_{cl} then the model after pre training? There is more material in the appendix on this but it would help the reader's understanding of this important concept of knowledge acquisition and knowledge loss to move some of that discussion into the body of the paper.
>
> The reviewer raises a question about lack of details in knowledge acquisition measurement.
> To evaluate the model's ability to acquire new knowledge while retaining pre-existing capabilities, we further train the intermediate checkpoints of OLMo within a continual knowledge learning setup. Here, $\theta_{pt}$​ represents the model at a specific stage of pretraining, defined by the number of steps completed during pretraining, while $\theta_{CL}$​ denotes the model after undergoing continual knowledge learning. The knowledge acquisition and retention capabilities of each model are assessed by comparing the performance differences between  $\theta_{CL}$ and $\theta_{pt}$.
> We will include additional details with examples in the next version to clear our explanation of the knowledge acquisition measurement method. We will also ensure the main text contains enough information for clarity.
>
> >Some minor stylistic corrections that will render the paper more readable:
>
> Thank you sincerely for providing such detailed and thoughtful feedback, even on minor aspects. We will look over the minor stylistic errors for the camera-ready version.
>
>
> **References**
>
> [1] Transformer feed-forward layers are key-value memories. Geva et al.,
>
> [2] OLMo: Accelerating the Science of Language Models Groeneveld et al.,
>
> [3] How Do Large Language Models Acquire Factual Knowledge During Pretraining Chang et al.,

---

### Official Review · Reviewer_zgzo · 2024-11-04

**Soundness:** 4
**Presentation:** 4
**Contribution:** 4
**Rating:** 8
**Confidence:** 4

**Summary:**

This work introduces the concept of "knowledge entropy" to analyze how language models integrate their stored knowledge throughout the pretraining process. By viewing feed-forward layers as key-value memory systems, the researchers measure how broadly or narrowly models access their stored knowledge by calculating the entropy of memory coefficients. Their key finding is that as pretraining progresses, models exhibit decreasing knowledge entropy, meaning they tend to rely on a narrower set of memory vectors rather than broadly integrating knowledge from many sources. This narrowing correlates strongly with decreased ability to acquire new knowledge and retain existing knowledge during continual learning, an interesting empirical finding.

**Strengths:**

The concept of "knowledge entropy" introduced in this paper is novel and very interesting. (1) Entropy based metrics have been used in previous works before. (2) Recent works have gradually built the view of feed-forward layers as key-value memory. This paper has a refreshed view, combining these 2 insights to give a new metric.

The "resuscitation" method, by manipulating parameters, presents somewhat a direct test for the authors' main claim that decreased knowlege entropy gives rise to decreased ability to learn new things.

**Weaknesses:**

This work only tested on OLMo models (1B and 7B), as acknowledged by authors. The authors state their reasons for this as due to the fact that OLMo provide intermediate checkpoints.

However the Pythia models, too, offer intermediate checkpoints. If time permits, the authors should endeavor to check their results on these models too, in order to ascertain the generality of their claims and whether it holds beyond a single model family.

It would also be useful if the authors provide some discussion regarding mechanisms behind their main finding (knowledge entropy correlates with decreased ability to acquire new knowledge). It does not need to be theorem based, which is a difficult endeavor. Empirical findings would help, to delve at least a bit deeper into this phenomenon and why it might occur.

**Questions:**

1. Why do you think that knowledge entropy correlates with decreased ability to acquire new knowledge?

2. Is the resuscitation method intended solely as a test for your main finding, or do you also envision a path in which this could practically be useful for increasing plasticity of models?

---

> ### Author Response · Authors · 2024-11-20
> **Response to Reviewer zgzo**
>
> Dear reviewer zgzo,
>
> The authors thank you sincerely for taking your valuable time to provide constructive feedback on this paper. We appreciate your acknowledging the novelty of our defined knowledge entropy, which combines the concept of entropy with key-value memory and the validity of the experiment to deliver our main claim. We present additional experiments and clarification regarding the reviewer’s concerns and questions below.
>
> > This work only tested on OLMo models (1B and 7B), as acknowledged by authors. The authors state their reasons for this as due to the fact that OLMo provide intermediate checkpoints. However the Pythia models, too, offer intermediate checkpoints. If time permits, the authors should endeavor to check their results on these models too, in order to ascertain the generality of their claims and whether it holds beyond a single model family.
>
> The reviewer raises a question about whether our findings are applicable to models beyond OLMo, such as Pythia. While we acknowledged that Pythia also offers intermediate checkpoints, we encountered challenges when analyzing its training dynamics. Specifically, Pythia models are trained on a much smaller dataset—approximately 10% of the OLMo training dataset. However, we observe that OLMo models show more stable training dynamics after about 15% of the training dataset has been processed, which is after the last checkpoint of Pythia.
>
> The early stages of training are influenced by a larger proportion of randomly initialized parameters, exhibit different behavior compared to models that have gone through more training steps and reached a stable state. Thereby, although Pythia provides the intermediate checkpoint, they all tend to be before reaching a stable state when analyzed with OLMo models, making Pythia not ideal for analyzing the training dynamics of language models. We believe this is one reason why previous work, such as [1] Chang et al. (2024) and [2] Sun and Dredz(2024), also focused on OLMo models for intermediate checkpoint analysis.
>
> As we cannot include figures in the rebuttal, we add the figure comparing the training dynamics of Pythia and OLMo in the revised version of our paper. Please refer to Figure 18 in the revised version of the paper for the result. Our analysis shows that while the overall training dynamics of Pythia are similar to those of OLMo, Pythia's early-stage training ends prematurely, before significant knowledge acquisition occurs. Additionally, we have included the performance of OLMo at 18k steps in our revised results to show the training dynamics before it stabilizes, which we estimate occurs at around 118k steps.

---

> ### Author Response · Authors · 2024-11-20
> **Response to Reviewer zgzo (2)**
>
> > It would also be useful if the authors provide some discussion regarding mechanisms behind their main finding (knowledge entropy correlates with decreased ability to acquire new knowledge). It does not need to be theorem based, which is a difficult endeavor. Empirical findings would help, to delve at least a bit deeper into this phenomenon and why it might occur.
>
> > Why do you think that knowledge entropy correlates with decreased ability to acquire new knowledge?
>
> The reviewer raises a question about the mechanisms behind the relationship between decreasing knowledge entropy and the ability to acquire and retain knowledge during pretraining. Our analysis is based on the intuition that the coefficients in the linear combination of memory vectors determine how the corresponding memory vectors are updated.
>
> As defined in Equation 1, the FFN output ($FFN(\mathbf{x}) = f(\mathbf{x} \cdot K^T) \cdot V$)  is a linear combination of memory vectors $\mathbf{v}_{i=1,\cdots,m} \in \mathbb{R}^{d}$
>
> (the row vectors of $V \in \mathbb{R}^{m \times d}$ ),
> where the coefficients $c_i$ are given by $f(\mathbf{x} \cdot K^T)$.  In other words, $FFN(x)=c_1\mathbf{v_1}+c_2\mathbf{v_2}+\cdots+c_m\mathbf{v_m}$. Within a given layer, since the operations beyond the FFN layer and the input to the FFN layer remain consistent across all memory vectors, the coefficients act as scaling factors for the gradient by the chain rule. During training, the gradient $\frac{ \partial L}{\partial \mathbf{v_{i,j}}}$ for $ i = 1, 2, \ldots, m$ and $ j = 1, 2, \ldots, d$ can be decomposed as: $$\frac{ \partial L}{\partial \mathbf{v_{i,j}}} = \frac{ \partial L}{\partial FFN(x)}\cdot \frac{ \partial FFN(x)}{\partial v_{i,j}}$$
>
> Here, $\frac{ \partial L}{\partial FFN(x)}$ is the same for all $\mathbf{v_i}$, meaning that the relative magnitude of the gradient depends on $\frac{ \partial FFN(x)}{\partial v_{i,j}} = c_i$. Thus, larger coefficients result in proportionally larger gradients being applied to the corresponding memory vectors, amplifying their updates during backpropagation.
>
> As pretraining progresses, a spikier coefficient distribution—captured by decreasing knowledge entropy—implies that gradient updates become increasingly concentrated on specific positions where the average coefficients are larger. This centralization can affect the model's ability to evenly utilize its memory capacity, impacting knowledge acquisition and retention.
>
> We included the intuition in our paper in Appendix A.1. We sincerely appreciate your valuable feedback.

---

> ### Author Response · Authors · 2024-11-20
> **Response to Reviewer zgzo (3)**
>
> > Is the resuscitation method intended solely as a test for your main finding, or do you also envision a path in which this could practically be useful for increasing plasticity of models?
>
> The reviewer asks whether the resuscitation method can be viewed as a practical method to increase the plasticity of models. While we observe that our resuscitation method reduces the forgetting rate and improves knowledge acquisition thereby supporting our main findings of the correlation between knowledge entropy and knowledge acquisition/forgetting rates, the primary motivation of the experiment was to test our hypothesis. As such, we did not explore alternative methods for manipulating model parameters and instead focused on a basic approach (Algorithm 1). Thereby, as noted in the paper, we observe that the end performance in knowledge acquisition and retention tends to be higher when selecting a middle-stage checkpoint with similar knowledge entropy, rather than achieving the same entropy through resuscitation. However, we view this approach as an initial attempt towards increasing model plasticity. We believe that refining the parameter manipulation approach beyond the simple technique used in Algorithm 1 could yield better overall performance, especially with larger q values, and think that this approach can be practically used to address major challenges in training language models with new knowledge in the future.
>
>
> **reference**
>
> [1] How Do Large Language Models Acquire Factual Knowledge During Pretraining? Chang et al.
>
> [2] Amuro & Char: Analyzing the Relationship between Pre-Training and Fine-Tuning of Large Language Models, Sun and Dredze.

---

> ### Author Response · Authors · 2024-11-25
> **Looking forward to discussions**
>
> Dear reviewer zgzo,
>
> We appreciate your time and effort in reviewing the paper.
>
> As we near the end of the discussion period, we would like to remind you of the upcoming deadline kindly. We are happy to discuss any aspects of the paper that may require further clarification.
>
> Thank you once again for your valuable feedback.

---

### Official Review · Reviewer_hpJw · 2024-11-04

**Soundness:** 2
**Presentation:** 2
**Contribution:** 2
**Rating:** 5
**Confidence:** 4

**Summary:**

The authors identified a phenomenon where language models use less diverse set of value vectors (i.e., the parametric memory store) to learn knowledge. The authors called the distribution "knowledge entropy" and hypothesize that the decrease in knowledge entropy hinders the models' ability to acquire and retain new knowledge. The authors found that the knowledge entropy decreases as pretraining progresses.

The authors further test the progression of knowledge acquisition (using fictional knowledge probes) and forgetting (using common knowledge datasets) with intervention on the up-projection matrix. The author resusitate the inactive memory vectors by finding the least active $p\%$ of the memory vectors and multiply the up-projection matrix by the ratio $q$ at the corresponding index. The authors show that this reduce forgetting and increase retention of new knowledge (I have questions regarding this -- see the question section).

**Strengths:**

This paper investigates an important problem of knowledge acquisition and retention in LLMs. The angle is simple, clear, and easy to validate. The idea is not particularly high on the originality end but the identified "variable" (i.e., the knowledge entropy) has not been used to understand forgetting/retention as far as I know.

**Weaknesses:**

**Main Issues**

The main concern is it's unclear how causal the knowledge entropy is to the phenomenon of knowledge forgetting/retention. The paper provided mostly correlational evidence. Please see the Questions section for more concrete critique and suggestions.

**Presentation Issues**

Quite a bit of notation overloading:
1. c is used for the memory coefficient in equation 2 and used again in section 4.1 L303 to represent knowledge.
2. $m$ is used for index in equation 2 and then as multiplier in algorithm 1
L320: missing space between sentences.

**Questions:**

In L209, the negative value is often understood to remove/erase the information from the residual stream. If this flexibility is removed from the model, it could affect the model's plasticity. It would be useful to see a more flexible gating function. The entropy can still be calculated using the absolute values of the coefficients (and does not need to change the model arch and training).

What does section 3.4 do to support the use of knowledge entropy? I understand the concept of attention entropy and next token entropy but how does that connect to the choice of using knowledge entropy? In fact, if all these entropy measures decrease over time, how can we justify that knowledge entropy is the main reason for the degraded plasticity even if it shows high correlation? Section 4.3 provides interventional evidence but similar intervention should be done with, say, attention entropy to ensure this is not the source for the lowered plasticity. I.e., if making attention entropy less peaky also increase retention and mitigates forgetting, then the role of knowledge entropy is not unique as other statistics reflect the same phenomenon. Would be nice to see more interventional evidence showing that knowledge entropy is the main contributor to the plasticity issue.

Figure 5 (b): seems that downstream performance decreases as the $q$ value gets larger, which is the opposite of knowledge retention? And the knowledge probing result in the same plot doesn't show improvement over lower $q$ values. Am I missing something or does it show a different trend than Figure 5 (a)?

I'm happy to increase my scores if the above questions can be addressed.

---

> ### Author Response · Authors · 2024-11-20
> **Response to Reviewer hpJw**
>
> Dear Reviewer hpJw,
>
> Thank you for your valuable time in reviewing our paper. We appreciate recognizing the importance of the problem we are investigating and the novelty of our analysis. We will fix the presentation issues in the final version.
>
> > In L209, the negative value is often understood to remove/erase the information from the residual stream. If this flexibility is removed from the model, it could affect the model's plasticity. It would be useful to see a more flexible gating function. The entropy can still be calculated using the absolute values of the coefficients (and does not need to change the model arch and training).
>
> The reviewer suggests calculating entropy using the absolute value of coefficients without altering the architecture or training. We would like to clarify that we did not change any architecture or training, but all is calculated the same during the forward pass. We apply absolute value to the calculated value $f(x · K^T )$ during normal forward pass only to calculate knowledge entropy, where f is the non-linear activation function, SwiGLU. We believe this is the same calculation as the reviewer suggested. Please let us know if we missed something.
>
> > What does section 3.4 do to support the use of knowledge entropy? I understand the concept of attention entropy and next token entropy but how does that connect to the choice of using knowledge entropy? In fact, if all these entropy measures decrease over time, how can we justify that knowledge entropy is the main reason for the degraded plasticity even if it shows high correlation? Section 4.3 provides interventional evidence but similar intervention should be done with, say, attention entropy to ensure this is not the source for the lowered plasticity. I.e., if making attention entropy less peaky also increase retention and mitigates forgetting, then the role of knowledge entropy is not unique as other statistics reflect the same phenomenon. Would be nice to see more interventional evidence showing that knowledge entropy is the main contributor to the plasticity issue.
>
> The reviewer raises questions about the connection between Section 3.4 and knowledge entropy and requests additional evidence to show that knowledge entropy, rather than attention entropy or next token entropy, is the primary factor contributing to reduced plasticity.
> We built the definition of knowledge entropy based on findings from [1]Geva et al. (2021), which demonstrated that the feed-forward layer captures knowledge. However, we acknowledge other works that explore the role of attention layers([2] Geva et al.(2023)) and knowledge circuits([3] Yao et al.(2024)). Thereby, in Section 3.4, we wanted to show those people who would be curious about the overall trend of the model apart from only the feed-forward layer that we observed that the overall model behavior exhibits similar trends across various entropy measures.
>
> Nevertheless, we focused on defining knowledge entropy specifically in the feed-forward layer because our findings suggest it is the primary factor in degraded plasticity. Our experiments with resuscitation targeting attention layers showed some improvements in knowledge acquisition and retention, but these effects were less pronounced compared to resuscitation in the feed-forward layer (details of experiment setting and results in next paragraph). This suggests that, while attention entropy and next token entropy may reflect certain trends, knowledge entropy associated with the feed-forward layer plays a more significant role in plasticity degradation.
>
> In the experiment where we introduce resuscitation into the attention layer, we apply temperature scaling to the dot product of key and query vectors, right before the softmax activation. While artificial interventions to increase Knowledge Entropy can be achieved by adjusting the parameters of the feed-forward layer at specific positions with low activation values, the same approach cannot be applied to alter Attention Entropy. This is because Attention Entropy reflects the probability distribution over prior token positions. To artificially increase Attention Entropy, we employed temperature scaling to reduce the sparsity of the probability distribution. We experimented with temperature values ranging from 1.5 to 3.0.
>
> As we are not able to attach the figure, we add the figure in Appendix B.7 of the revised paper (Figure 17). The figure shows the results of resuscitation performed in the feed-forward and attention layers under varying temperature settings. The data indicates that resuscitation in the feed-forward layer consistently yields the highest knowledge acquisition rate and generally achieves the highest knowledge retention rate. This trend is reflected in both knowledge probe results and downstream performance. These findings suggest that Knowledge Entropy plays a critical role in influencing knowledge acquisition and forgetting, driving the observed differences in performance.

---

> > ### Author Response · Authors · 2024-11-20
> > **Response to Reviewer hpJw**
> >
> > >Figure 5 (b): seems that downstream performance decreases as the q value gets larger, which is the opposite of knowledge retention? And the knowledge probing result in the same plot doesn't show improvement over lower q  values. Am I missing something or does it show a different trend than Figure 5 (a)?
> >
> > The reviewer raises a question about the interpretation of the plot in Figure 5(b) compared to Figure 5(a), specifically regarding the decrease in downstream performance as the q value increases and the lack of improvement in knowledge probing results for higher q values (We believe you may have meant "higher "q instead of "lower q" regarding knowledge probing result but if this is incorrect, please let us know.).
> >
> > Figure 5(a) shows the improvement and degradation rate of knowledge acquisition and forgetting, respectively, which represent the *difference relative to the initial state* (equation in Appendix B.2). Importantly, the initial performance varies with q due to the scaling effect introduced in Algorithm 1. This means that even with lower forgetting rates, downstream performance may not always increase for larger q values because the initial performance for different q values is not the same.
> >
> > In Figure 5(b), we observed that for larger q values, the higher acquisition rate and lower forgetting rate generally led to better knowledge probing and downstream performance compared to the original state. For the case of q=3, as the parameter manipulation get more severe with larger q leading to a larger initial performance drop, we could see even with a low forgetting rate, it resulted in lower downstream performance than the original.
> >
> > We would like to emphasize that the main goal of the resuscitation experiment was to investigate whether changes in knowledge entropy drive the differences in knowledge acquisition and forgetting rates, while keeping the same parameter state and altering knowledge entropy using Algorithm 1. As shown in Figure 5(a), increasing the q value leads to higher knowledge acquisition and forgetting rates, supporting our hypothesis.
> >
> > **References**
> >
> > [1] Transformer feed-forward layers are key-value memories. Geva et al.,
> >
> > [2] Dissecting recall of factual associations in auto-regressive language models. Geva et al.,
> >
> > [3] Knowledge circuits in pretrained transformers. Yao et al.,

---

> ### Author Response · Authors · 2024-11-25
> **Looking forward to discussion**
>
> Dear Reviewer hpJw,
>
> Thank you for your time and effort in reviewing our paper.
>
> As the discussion period approaches its conclusion, we would like to kindly remind you of the upcoming deadline. We are happy to discuss any aspects of the paper that require further clarification or discussion.
>
> Once again, we sincerely appreciate your valuable feedback and insights.

---

> ### Author Response · Authors · 2024-11-28
>
> Dear Reviewer hpJw,
>
> Thank you for your effort in reviewing our paper. We have addressed the notation overload issue you pointed out and corrected grammatical errors, revising the paper accordingly. Additionally, we have included an intuition about the relationship between decaying knowledge entropy and knowledge acquisition and retention performance in Appendix A.1 of the revised paper. We hope this revision could alleviate your concerns related to causality. It would be appreciated if we could discuss any remaining concerns during the extended review period.

---

### Meta-Review · Area_Chair_TJmB · 2024-12-11

**Metareview:**

This paper introduces the concept of "knowledge entropy" to analyze how language models store and access knowledge and present empirical evidence showing that knowledge entropy decreases as models are trained, correlating with a reduced ability to learn new information and an increased tendency to forget existing knowledge.

This is a novel approach that provides useful insights to an relevant problem in LM pre-training. The experiments are novel / creative and well supported. Reviewers agree that this is a valuable addition to the literature.

Weaknesses like notation overload and relation to previous work were addressed in rebuttal.

**Additional Comments On Reviewer Discussion:**

Weaknesses like notation overload and relation to previous work were addressed in rebuttal. Decent overall engagement. One mildly negative reviewer did not engage in discussion at all.

---

### Decision · Program_Chairs · 2025-01-22

Accept (Oral)